# DIFFREE: TEXT-GUIDED SHAPE FREE OBJECT IN-PAINTING WITH DIFFUSION MODEL

## ABSTRACT

This paper addresses an important problem of object addition for images with only text guidance. It is challenging because the new object must be integrated seamlessly into the image with consistent visual context, such as lighting, texture, and spatial location. While existing text-guided image inpainting methods can add objects, they either fail to preserve the background consistency or involve cumbersome human intervention in specifying bounding boxes or user-scribbled masks. To tackle this challenge, we introduce Diffree, a Text-to-Image (T2I) model that facilitates text-guided object addition with only text control. To this end, we curate OABench, an exquisite synthetic dataset by removing objects with advanced image inpainting techniques. OABench comprises 74K real-world tuples of an original image, an inpainted image with the object removed, an object mask, and object descriptions. Trained on OABench using the Stable Diffusion model with an additional mask prediction module, Diffree uniquely predicts the position of the new object and achieves object addition with guidance from only text. Extensive experiments demonstrate that Diffree excels in adding new objects with a high success rate while maintaining background consistency, spatial appropriateness, and object relevance and quality.

## 1 INTRODUCTION

With the recent remarkable success of Text-to-Image (T2I) models (e.g., Stable Diffusion (Podell et al., 2023), Midjourney (Midjourney, 2022), and DALL-E (Betker et al., 2023; Ramesh et al., 2022)), creators can quickly generate high-quality images with text guidance. The rapid development has driven various text-guided image editing techniques (Brooks et al., 2023; Geng et al., 2023; Zhang et al., 2024; 2023; Sheynin et al., 2023). Among these techniques, text-guided object addition which inserts an object into the given image has attracted much attention due to its diverse applications, such as advertisement creation, visual try-on, and renovation visualization. While important, object addition is challenging because the object must be integrated seamlessly into the image with consistent visual context, such as illumination, texture, and spatial location.

Existing techniques for object addition in images can be broadly categorized into mask-guided and text-guided approaches, as depicted in Figure 2. Mask-guided algorithms typically require the specification of a region where the new object will be inserted. For example, traditional image inpainting methods (Bertalmio et al., 2000; Suvorov et al., 2022; Lugmayr et al., 2022; Yu et al., 2018; Pathak et al., 2016) focus on seamlessly filling user-defined masks within an image to match the surrounding context. Recent advancements, such as PowerPaint (Zhuang et al., 2024), have effectively incorporated objects into images given their shape and textual descriptions while maintaining background consistency. However, manually delineating an ideal region for all objects, considering shape, size, and position, can be labor-intensive and typically requires drawing skills or professional knowledge. On the other hand, text-guided object addition methods, such as InstructPix2Pix (Brooks et al., 2023), attempt to add new objects using only text-based instructions. Despite this, these methods have a low success rate and often result in background inconsistencies, as demonstrated in Figure 2 and Figure 7. Additionally, when employing text-guided methods for iterative object addition, the quality of the inpainted image tends to degrade progressively with each step, as depicted in Figure 8.

To tackle the above challenges, we introduce Diffree, a diffusion model with an additional object mask predictor module that can predict an ideal mask for a candidate inpainting object and achieve

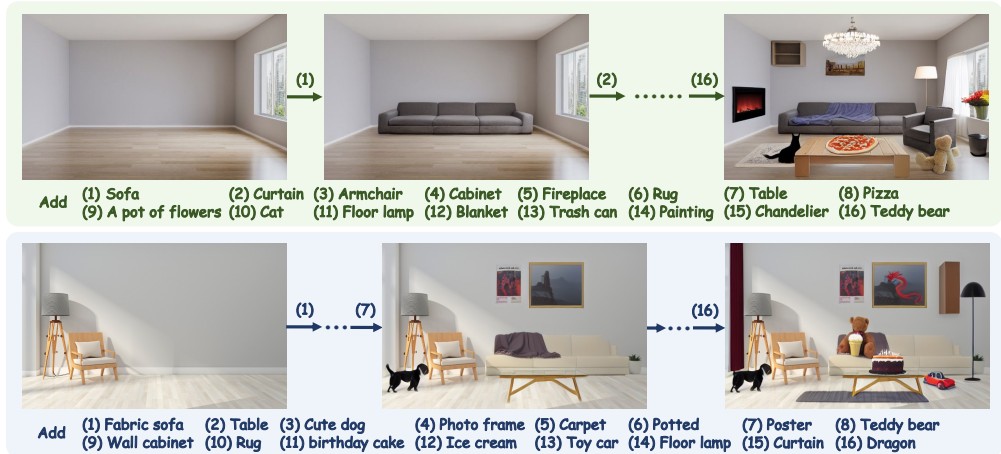

Figure 1: Diffree iteratively generates object additions, ensuring text-guided objects are reasonably added while maintaining background consistency. Refer to Figure A1 for the complete process.

shape-free object addition with only text guidance. Compared with previous works (Xie et al., 2023; Zhuang et al., 2024; Brooks et al., 2023), our Diffree has three appealing properties. *First*, Diffree can achieve impressive text-guided object addition results while keeping the background unchanged. In contrast, previous text-guided methods (Brooks et al., 2023) struggle to guarantee this. *Second*, Diffree does not require additional mask input, which is necessary for traditional mask-guided methods (Xie et al., 2023). In real scenarios, high-quality masks are hard to obtain. *Third*, Diffree can generate the instance mask, helping prevent quality degradation of iterative addition (i.e., Figure 8) or can be used to combine with various existing works to develop exciting applications. For example, Diffree can achieve image-prompted object addition when combined with AnyDoor (Chen et al., 2023) and plan to add objects suggested by GPT4V (OpenAI, 2023), as shown in Figure 9.

Towards high-quality text-guided object addition, we curate a synthetic dataset named Object Addition Benchmark (OABench) which consists of 74K real-world tuples including an original image, an inpainted image, a mask image of the object, and an object description. The data curation process is illustrated in Figure 5. Note that object addition can be deemed as the inverse process of object removal. We build OABench by removing objects in the image using advanced image inpainting algorithms such as PowerPaint (Zhuang et al., 2024). In this way, we can obtain an original image containing the object, an inpainted image with the object removed, the object mask, and the object descriptions. We use instance segmentation dataset COCO (Gupta et al., 2019; Lin et al., 2014) as the source data, which has two benefits. First, the source image captures comprehensive natural scenes where the location and shape of one individual object often exhibit intrinsic alignment with the overall scene. It helps guarantee the reasonability of new objects' location. For instance, a monitor is commonly situated behind computer peripherals. Second, the ground-truth mask of the object already exists in the instance segmentation dataset, which can be directly utilized to remove objects with background consistency preserved. By contrast, InstructPix2Pix (Brooks et al., 2023) collects image pairs using proprietary T2I model (Rombach et al., 2022) under prompt pair with subtle modifications. While this approach maintains new objects' reasonability, it poses difficulties in preserving background consistency.

With OABench, Diffree is trained to predict masks and images containing the new object given the original image and object description. Thanks to the extensive coverage of objects in natural scenes in OABench, Diffree can add various objects to the same image while matching the visual context well as shown in Figure 3. Moreover, Diffree can iteratively insert objects into a single image while preserving the background consistency using the generated mask as shown in Figures 1 and 4.

For evaluation, we propose a set of evaluation rules through existing metrics (Hessel et al., 2021; Zhang et al., 2018; Heusel et al., 2017; Xie et al., 2023; OpenAI, 2023), including consistency of background, reasonableness of object location, quality, diversity and correlation of generated object, and success rate. Extensive experiments show that Diffree performs better in object addition than previous mask-guided and text-guided techniques. For instance, Diffree obtains a significantly higher success rate than InstructPix2Pix. For successful cases, Diffree still outperforms Instruct-Pix2Pix in various quantitative metrics.

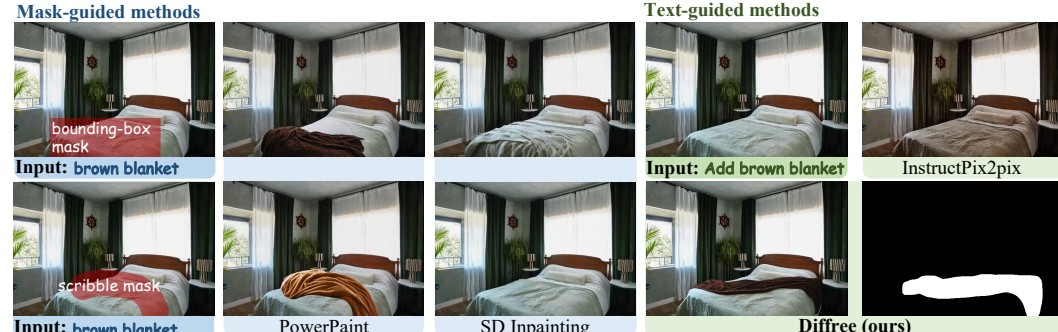

Figure 2: Qualitative comparisons of Diffree and various other methods.

The **contributions** of this work are three-fold. 1) We proposed Diffree, a model that can achieve text-guided shape-free object addition to free users from drawing the appropriate mask of objects. The inpainted image from Diffree includes the new objects with reasonable shapes and consistent visual context. 2) We introduced OABench, an exquisite synthetic dataset for object addition. OABench comprises 74K real-world training data for the task of object addition. 3) We evaluate this task with a set of rules for comprehensive assessment. Extensive experiments demonstrate the effectiveness of Diffree. For example, Diffree achieves a high success rate (e.g., 98.5% in COCO) and superior unified score (e.g., 38.92 versus 4.48) compared with other methods.

## 2 RELATED WORK

**Text-to-Image Diffusion Models** Recently, text-to-image (T2I) diffusion models (Nichol et al., 2022; Ramesh et al., 2022; Betker et al., 2023), have shown exceptional capability in image generation quality and extraordinary proficiency in accurately following text prompts, under the dual support of large-scale text-image dataset (Schuhmann et al., 2022; Zhao et al., 2024) and model optimizations (Dhariwal & Nichol, 2021; Ho et al., 2020; Rombach et al., 2022). DALLE-2 (Ramesh et al., 2022) enhances text-image alignment via CLIP (Radford et al., 2021) joint feature space, DALLE-3 (Betker et al., 2023) further improves the prompt following abilities by training on highly descriptive generated image captions. Stable Diffusion (Rombach et al., 2022), which is well-established and widely adopted, garners significant attention and application within and beyond the research community. Given that T2I models generate comprehensive images from text prompts, even minor alterations in prompts can result in substantial changes to the resultant image (Brooks et al., 2023). Consequently, there has been an increased focus not only on T2I generation but also on image editing based on additional conditions such as text inputs, masks, et al.

**Text-Guided Image Editing** The effectiveness of the text-guided image editing methods (Brooks et al., 2023; Zhang et al., 2024; Sheynin et al., 2023) largely depends on the composition of its dataset and how it is collected. InstructPix2Pix (Brooks et al., 2023) combines two large pretrained models, a large language model (Mann et al., 2020) and a T2I model (Rombach et al., 2022), to generate a dataset for training a diffusion model to follow written image editing text prompts. Its innovative data collection method allows InstructPix2Pix to follow instructions and shows amazing effects while making its consistency difficult to guarantee due to both input and output being generated by the T2I model. Emu Edit (Sheynin et al., 2023) adapts its architecture for multi-task learning by framing an extensive array of tasks as generative tasks, demonstrating robust and versatile performance. MagicBrush (Zhang et al., 2024) introduces a manually annotated dataset in which the T2I model generates both input and output. The image editing performance of fine-tuning InstructPix2Pix on MagicBrush shows better. Unlike the previous methods, we propose a novel and easily expandable collection method, thanks to the existing instance segmentation dataset, we use real images as output and synthetic images without a specific object as input. SmartMask Singh et al. (2024) predicts masks for added objects but relies on additional segmentation and mask-guided inpainting models, along with detailed scene descriptions. In contrast, our approach uses a single model with an object description, eliminating complexity, resource dependency, and potential limitations stemming from external models or detailed input requirements. Our work closely relates to the concurrent work PIPE (Wasserman et al., 2024), which independently explores similar concepts and methodologies. Both studies involve removing objects to collect an object addition dataset and train a diffusion model for text-guided object addition. Our approach additionally trains an Ob-

Figure 3: Diffree adds objects to the same image, with different spatial relationships.

ject Mask Predictor (OMP) module to predict the mask of objects. We believe that the concurrent exploration of these ideas underscores the significance and timeliness of this research direction.

**Mask-Guided Image Inpainting** Mask-guided image inpainting methods (Xie et al., 2023; Zhuang et al., 2024; Chen et al., 2023) alter the image in specific areas under additional conditions (e.g., text), while maintaining the background in its original state. SmartBrush (Xie et al., 2023) achieves precise object inpainting guided by text and mask through a novel training and sampling strategy. AnyDoor (Chen et al., 2023) employs a discriminative ID extractor and a frequency-aware detail extractor to characterize the target object, thereby facilitating effective object addition given an area and corresponding object image. Powerpaint (Zhuang et al., 2024) demonstrates superior performance on various inpainting benchmarks attributed to introducing learnable tokens to distinguish different tasks. Although these methods have achieved amazing image inpainting effects, their commonality is the need for a mask. For ordinary users, drawing an object mask with an appropriate shape, size, and aspect ratio, corresponding accurately to the object and image, presents an unignorable challenge. Certain mask-guided methods (Nichol et al., 2022; Li et al., 2023) eschew the need for precise mask conditions, utilizing instead approximate masks (e.g., Glide (Nichol et al., 2022)) or bounding boxes (e.g., GLIGen (Li et al., 2023)). While these approaches relax constraints on specific shapes, they still necessitate the specification of reasonable size and position, thereby introducing challenges, as discussed in Section A7 of the Appendix.

## 3 METHODOLOGY

Given an image and the object description, our goal is to add the object to the image while preserving the background consistency. Following this, we initially introduce OABench, a synthetic dataset for this task, comprising image-text pairs with corresponding object masks and images containing the object. We provide an overview of our data collection pipeline in Section 3.1. We next present Diffree, an architecture amalgamating a Stable Diffusion model with an Object Mask Predictor (OMP) module in Section 3.2. The evaluation procedure is presented in Section 3.3.

### 3.1 OABENCH

We combine existing instance segmentation dataset (Gupta et al., 2019; Lin et al., 2014) with powerful image inpainting method (Zhuang et al., 2024) to generate the OABench. Unlike other instructions follow methods (Brooks et al., 2023; Zhang et al., 2024), generating both data pairs using existing text-to-image (T2I) models (Rombach et al., 2022; Ramesh et al., 2022) with prompt pairs and filtering, we use the real image with objects to synthesize the image without the object, as depicted in Figure 5. Furthermore, an object in the real image naturally aligns with its background, i.e., it is appropriate for generating the corresponding image without the same object. The tri-phase data generation process is described in the following sections, with comprehensive procedural specifics detailed in Section A5 of the Appendix.

**Collection and Filtering** We gather and refine instances suitable for image inpainting by applying a set of rules from the LVIS dataset (Gupta et al., 2019), a large instance segmentation dataset annotated for COCO (Lin et al., 2014) dataset. As depicted in Figure 5, in images containing multiple instances, we enforce size constraints to exclude instances that are too big or too small (typically related to object components or background elements like buttons on clothing or rivers). Subsequently,

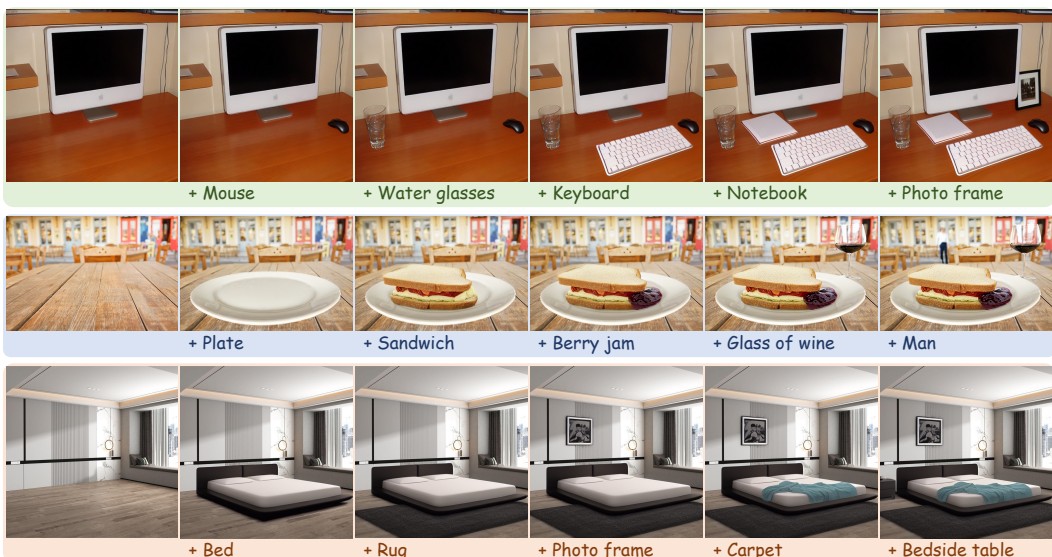

Figure 4: Diffree iteratively generates results. Objects added later can relate to the earlier.

incomplete instances are filtered out using edge detection and integrity assessments. Instances that are partially obscured are identified through cavity inspection, iterative IOU algorithm application, and common part comparison among various instances. Additionally, objects with exceptionally high aspect ratios, which tend to yield subpar inpainting outcomes, are also eliminated.

**Data Synthesis** We next employ a powerful image inpainting method, PowerPaint (Zhuang et al., 2024), to eliminate specific instances obtained in the preceding stage. Therefore, we can generate a synthetic image without specific objects with background consistency with the original image. Simultaneously, the object mask and corresponding name can be extracted from the LVIS and COCO.

**Post-Processing** In the post-processing stage, we filter out the results with poor effects in image inpainting. For some special cases (e.g., one of many dense and adjacent small cakes), image inpainting cannot effectively remove objects due to the complexity of the background. Thus we calculate the clip score (Hessel et al., 2021) using the object name and the region of the inpainted image, setting a threshold to remove images with higher scores that are deemed suboptimal. Finally, OABench includes 74,774 high-quality data pairs, each data pair includes a synthetic image and object caption as input, object masks and original images as output.

## 3.2 DIFFREE

For an image $x$ and a text prompt $d$, Diffree predicts a binary mask $m$ that specifies the region in $x$ and generates an image $\tilde{x}$. The masked region $\tilde{x} \odot m$ aligns with the text prompt $d$. To this end, Diffree is instantiated with a pre-trained T2I diffusion model (e.g. Stable Diffusion (Rombach et al., 2022)) with an object mask prediction (OMP) module as shown in Figure 6.

**Diffusion Model** learns to generate data samples by iteratively applying denoising autoencoders that estimate the score function (Song et al., 2020) of a given data distribution (Sohl-Dickstein et al., 2015). Stable Diffusion (Rombach et al., 2022) apply them in the latent space of powerful variational autoencoder (Kingma & Welling, 2013), including encoder $\mathcal{E}$ and decoder $\mathcal{D}$, to reduce computing resources while maintaining quality and flexibility. Stable Diffusion encompasses both forward and reverse processes. Given an image $\tilde{x}$, the forward process adds noise to the encoded latent $\tilde{z} = \mathcal{E}(\tilde{x})$:

$$\tilde{z}_t = \sqrt{\bar{\alpha}_t}\tilde{z} + \sqrt{1 - \bar{\alpha}_t}\epsilon, \epsilon \sim \mathcal{N}(0, \mathbf{I}) \tag{1}$$

where $\tilde{z}_t$ is the noisy latent at timestep $t$, $\bar{\alpha}_t$ denotes the associated noise level.

In the reverse process, we learn a network $\epsilon_\theta$ that predicts the noise added to the noisy latent $\tilde{z}_t$, conditioned on both the image $x$ and text $d$. To fine-tune Stable Diffusion for inpainting, we extend the channel of the first convolution layer to concatenate latent $z = \mathcal{E}(x)$ of image $x$ with $\tilde{z}_t$. This allows Diffree to generate images by denoising step by step from Gaussian noise concatenated with

Figure 5: The data collection process of OABench.

the latent of the input image. At the same time, the denoising process is guided by the associated feature $\text{Enc}_{\text{txt}}(d)$ of text $d$ encoded through the CLIP text encoder (Radford et al., 2021). The network $\epsilon_\theta$ is optimized by minimizing the following objective function:

$$L_{\text{DM}} = \mathbb{E}_{\mathcal{E}(\tilde{x}), \mathcal{E}(x), d, \epsilon \sim \mathcal{N}(0, \text{I}), t}\left[\|\epsilon - \epsilon_\theta(\tilde{z}_t, z, \text{Enc}_{\text{txt}}(d), t)\|_2^2\right]. \tag{2}$$

**OMP Module** and diffusion model are trained simultaneously and used to predict the binary mask $m$. The OMP module, which maintains a generally symmetric structure, comprises two convolutional layers, two ResBlocks, and an attention block, as illustrated in Figure 6. First, we calculate the predicted noise-free latent $\tilde{o}_t$ using the output of the diffusion model:

$$\tilde{o}_t = \frac{\tilde{z}_t - \sqrt{1 - \bar{\alpha}_t}\epsilon_\theta(\tilde{z}_t, z, \text{Enc}_{\text{txt}}(d), t)}{\sqrt{\bar{\alpha}_t}}. \tag{3}$$

Here, the concatenation of $z = \mathcal{E}(x)$ with $\tilde{o}_t$ serves as inputs to the OMP module. The gradient of $\tilde{o}_t$ is detached to optimize the two models without affecting each other. We conduct bilinear interpolation downsampling on the mask $m$ to obtain $m'$, preserving its size identical to the input latent. The OMP module's network $\tau_\theta$ is optimized according to the following objective function:

$$L_{\text{OMP}} = \mathbb{E}_{\mathcal{E}(\tilde{x}), \mathcal{E}(x), d, m}\left[\|m' - \tau_\theta(\tilde{o}_t, z)\|_2^2\right]. \tag{4}$$

It is noteworthy that the OMP module can predict the mask through the reverse process of diffusion rather than after it, as $\tilde{o}_t$ is available at each step, enabling mask prediction in the initial steps, as illustrated in Figure A4 in the Appendix. We train both the diffusion model and the OMP module simultaneously. Combining Equations (2) and (4), our final training objective can be expressed as:

$$L = L_{\text{DM}} + \lambda L_{\text{OPS}}, \tag{5}$$

where $\lambda$ is a hyper-parameter which balances the two losses.

**Classifier-free Guidance** Classifier-free diffusion guidance (Ho & Salimans, 2022) is a method that involves the joint training of a conditional diffusion model and an unconditional diffusion model. By combining the output score estimates from both models, this approach achieves a balance between sample quality and diversity. Training for the unconditional diffusion model is achieved by fixing the conditioning value to a null variable intermittently throughout the training process. We follow the approach of Brooks et al. (2023) by stochastically and independently defining our input conditions $x$ and $d$ as null variables with a probability of 5%.

### 3.3 EVALUATION METRIC

Due to the absence of robust quantitative metrics for shape-free object inpainting except the success rate, we propose a set of evaluation rules leveraging exits metrics (Hessel et al., 2021; Zhang et al., 2018; Heusel et al., 2017; OpenAI, 2023) to evaluate different methods in different aspects.

We first randomly select and manually inspect 1,000 evaluation data pairs from COCO (Lin et al., 2014) and OpenImages (Kuznetsova et al., 2020) independently to ensure the validity of the object in the image and generalizability of the evaluation dataset. Each data pair comprises an original

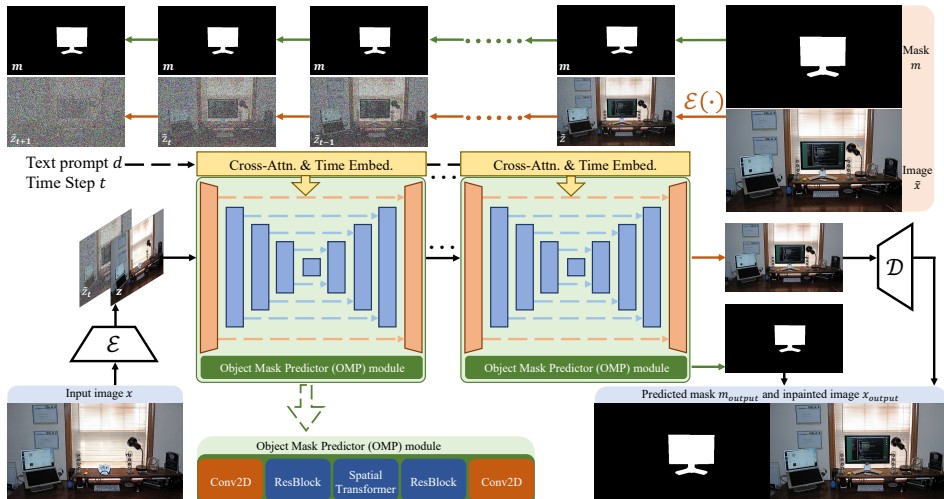

Figure 6: Diffree framework overview: The framework predicts the synthetic image with added objects and their masks. During training, diffusion model takes a concatenation of the input latent $z$ and the noisy output latent $\tilde{z}_t$ to estimate the noise at each timestep. The estimated noise is then used to denoise $\tilde{z}_t$, producing a noise-free latent $\tilde{o}_t$ (Equation (3)). The concatenation of $z$ and $\tilde{o}_t$, including input and denoised output information, is passed to OMP to generate the object mask $m$.

image $x_{\text{ori}}$, a text prompt of an object $d$, and an inpainted image $x$. The resulting output image $x_{\text{output}}$ and the corresponding object mask $m_{\text{output}}$ are outcomes derived from distinct methods.

**Background Consistency** We adapt LPIPS (Zhang et al., 2018), a widely adopted and robust metric for assessing the similarity between images, to evaluate this aspect:

$$s_{\text{con}}(x, x_{\text{output}}, m_{\text{output}}) = \text{LPIPS}(x, x \odot m_{\text{output}} + x_{\text{output}} \odot (1 - m_{\text{output}})). \quad (6)$$

**Location Reasonableness** Assessing the object's location's reasonableness is challenging due to its inherent subjectivity. Surprisingly, we note GPT4V (OpenAI, 2023) demonstrates strong discriminative abilities in assessing variations and evaluating different locations by providing $x$, $d$, $x_{\text{output}}$ and an instruction $T$ as illustrated in Figure A5 in the Appendix. GPT4V rates the appropriateness of the object's position on a scale from 1 to 5, while also providing justifications for these ratings:

$$s_{\text{rea}}(x, x_{\text{output}}, d, T) = \text{GPT4V}(x, x_{\text{output}}, d, T) \quad (7)$$

**Object Correlation** To quantify this relationship, we utilize CLIP Score (Hessel et al., 2021), a metric to assess the correlation between text and image, by calculating the cosine similarity of their embeddings from CLIP (Radford et al., 2021). we measure CLIP Score between the object area of $x_{\text{output}}$ and $d$, which is referred to as "Local CLIP Score":

$$s_{\text{cor}}(d, x_{\text{output}}, m_{\text{output}}) = \text{CLIPScore}(d, \text{Local}(x_{\text{output}}, m_{\text{output}})). \quad (8)$$

where $\text{Local}(x, m)$ denotes obtaining a cropped region from $x$ using $m$. To mitigate influences from background or mask shape, we compute an average of two Local CLIP Scores (one with background removal and another without).

**Object Quality and Diversity** Following (Xie et al., 2023), we employ Local FID, measuring Fréchet Inception Distance (FID) (Heusel et al., 2017) on the local regions, to evaluate the quality and diversity of generated object:

$$s_{\text{qd}}(LX_{\text{org}}, LX_{\text{output}}) = ||\mu_{LX_{\text{org}}} - \mu_{LX_{\text{output}}}||^2 +$$
$$\text{Tr}(\Sigma_{LX_{\text{org}}} + \Sigma_{LX_{\text{output}}} - 2 * (\Sigma_{LX_{\text{org}}} * \Sigma_{LX_{\text{output}}})^{\frac{1}{2}}) \quad (9)$$

where $LX_{\text{org}}$ and $LX_{\text{output}}$ respectively denote the sets comprising all local regions of the original images and output images, $\mu$ and $\Sigma$ represent the mean and variance of the feature vectors obtained through a particular network (Heusel et al., 2017).

**Unified Metric** Drawing upon the evaluation metrics delineated above (Equations (6) to (9)), we compute a unified score to holistically assess text-guided shape-free object inpainting. We treat the

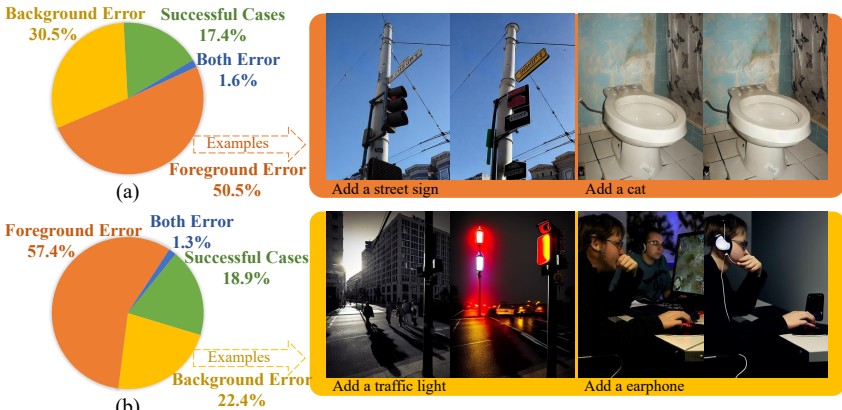

Figure 7: InstructionPix2Pix exhibits a low success rate in object addition, achieving 17.4% on (a) COCO and 18.9% on (b) OpenImages. Foreground Error refers to the failure in adding new objects or mistransforming existing ones, while Background Error indicates background inconsistencies.

derivative of inverse metric results (LPIPS and Local FID) as positive metrics and normalized the outcomes across different methods for each metric. Ultimately, we average these normalized scores and multiply them by the success rate as a unified score. The Unified metric not only considers success rate but also focuses on quantitative performances.

## 4 EXPERIMENT

We comprehensively evaluated our model, Diffree, by conducting experiments on two benchmark datasets: COCO (Lin et al., 2014), and OpenImages (Kuznetsova et al., 2020). Given the distinct input-output characteristics of our method compared to previous approaches, a quantitative comparison proves challenging. We align previous methods by adding auxiliary conditions, as depicted in Section 4.1, and provide quantitative comparison results (Section 4.2) to prove the effectiveness of Diffree more intuitively. We then showcase visualizations of generated images and give corresponding analyses to offer an intuitive assessment of Diffree's capabilities and comparisons in Section 4.3. Finally, we demonstrate some applications to prove that Diffree is highly compatible with existing methods (Section 4.4). Limitations and failure cases of Diffree are discussed in Section A4 of the Appendix, with further comparisons to other methods provided in Section A7 of the Appendix.

### 4.1 EXPERIMENTAL SETTINGS

**Training Setups** we employ OABench to train Diffree, initializing the diffusion model with the Stable Diffusion 1.5 (Rombach et al., 2022) weights. We set $\lambda = 2$ in Equation (5) and set a batch size of 64. Our model was trained around 10K steps on 4 A100 GPUs.

**Evaluation Datasets and Metrics** As outlined in Section 3.3, we employ LPIPS (Zhang et al., 2018), GPT4V Score, Local CLIP Score and Local FID (Xie et al., 2023) alongside the unified metric to assess performance on COCO (Lin et al., 2014) and OpenImages (Kuznetsova et al., 2020).

**Baselines** To facilitate comparison with prior methods (Brooks et al., 2023; Zhuang et al., 2024), we manually check and annotate the object masks for InstructPix2Pix (Brooks et al., 2023) (text-guided method with a low success rate, only considering successful instances), and utilize Diffree's mask output to assist PowerPaint (Zhuang et al., 2024) generation (mask-guided method). It is important to note that neither of these methods can complete evaluations independently. Thus, their quantitative metrics should be used as references only.

### 4.2 MAIN RESULTS

Table 1 shows the main results of Diffree with baselines. We report the results of four powerful metrics and a Unified Metric. It is worth highlighting that *only successful cases of InstructPix2pix are computed for these four metrics and PowerPaint (Mask-guided method) is utilized for image inpainting under the masks provided by our approach Diffree*.

Table 1: Main results on COCO and OpenImages. *: only calculate the successful cases' results. †: use the masks from our Diffree as PowerPaint's input.

| | | InstructPix2pix (Brooks et al., 2023) | PowerPaint (Zhuang et al., 2024) | Diffree (Ours) |
|---|---|---|---|---|
| | Success rate | 17.4 | N/A | **98.5** |
| COCO (Lin et al., 2014) | LPIPS ↓ | 0.11* | **0.06** | 0.07 |
| | GPT4V Score ↑ | 3.13* | N/A | **3.47** |
| | Local CLIP Score ↑ | 29.30* | 28.74 | **28.96** |
| | Local FID ↓ | 156.25* | 58.08 | **57.43** |
| | Unified Metric ↑ | 4.48 | 37.20† | **35.92** |
| | Success rate | 18.9 | N/A | **98.0** |
| OpenImages (Kuznetsova et al., 2020) | LPIPS ↓ | 0.11* | **0.06** | 0.07 |
| | GPT4V Score↑ | 3.36* | N/A | **3.50** |
| | Local CLIP Score ↑ | 29.21* | 28.57 | **28.81** |
| | Local FID ↓ | 143.82* | 62.40 | **60.07** |
| | Unified Metric ↑ | 5.04 | 36.41† | **35.47** |

**Success Rate** We achieved a success rate of over 98% on different datasets, while InstructPix2pix shows a lower success rate in object addition (17.2% and 18.9%). As shown in Figure 7, most of the results of InstructPix2pix involve replacing existing objects, without adding or significant changes to the background. This demonstrates our excellent ability to complete this task. Meanwhile, it is not applicable to PowerPaint as it necessitates a mask input.

**Consistency of Background** Diffree significantly outperforms InstructPix2pix in the LPIPS scores across all datasets (all decreased by 36% than InstructPix2pix). In particular, only scores from carefully chosen successful cases of InstructPix2pix were computed, potentially leading to an overestimation. Furthermore, Diffree, as a shape-free inpainting method, yields LPIPS results comparable to PowerPaint, as a shape-required inpainting method. As discussed in Section 3.1, we expect to achieve consistency of background like the image inpainting methods that necessitate masks. These methods inherently excel in this aspect, given that their input and ground truth are the same image during the training process. Therefore, we believe that we have a strong capability in this aspect.

**Reasonableness of object location** The results of GPT4V's assessment demonstrate that Diffree has a considerable advantage in the reasonableness of object location (e.g., 0.34 higher than only successful cases from InstructPix2pix). This is not available for PowerPaint due to it requires object location through a mask. We additionally present user study results in Figure A6 in the Appendix.

**Correlation, Quality and Diversity of Generated Object** We evaluate the generated object across these three dimensions, utilizing both Local CLIP Score and Local FID. Although Diffree exhibits a slightly lower Local CLIP Score in comparison to InstructPix2pix (e.g., 28.96 versus 29.30 on the COCO), this discrepancy can be rationalized by the fact that its successful results are inherently highly correlated while ours encompass all outcomes without any specific selection. Intriguingly, we demonstrate superiority over PowerPaint in terms of correlation. Furthermore, our performance according to the Local FID metric indicates a distinct advantage relative to all other methods.

**Unified Metric of Diffree** We combine the success rate with diverse metrics across various aspects to calculate a unified metric, thereby facilitating a more comprehensive comparison with extant text-guided methods. It is discernible that Diffree exhibits a substantial superiority over InstructPix2pix, for instance, ours' 35.92 as opposed to InstructPix2pix's 4.48 on the COCO. PowerPaint achieves superior results (e.g., 37.20 on the COCO dataset), however, a necessary input condition for its performance is the masks generated by our Diffree model. This further underscores the excellent scalability of Diffree when integrated with other methods.

## 4.3 VISUALIZATION

We provide different types of visualizations to more intuitively evaluate Diffree's capabilities Figures 1 to 4, 8 and 9, please refer to the respective image captions for detailed explanations. *For additional visualization results, please refer to the Section A1 of the Appendix.*

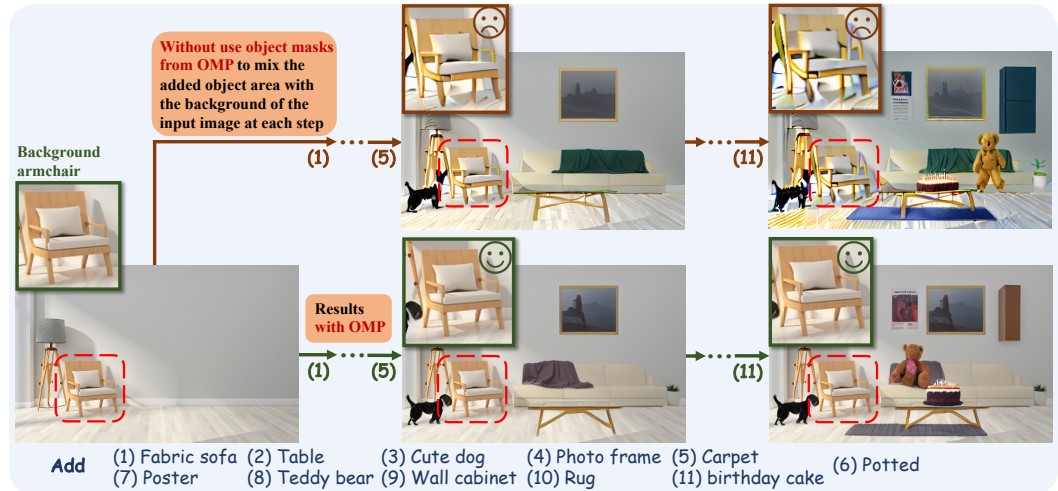

Figure 8: Ablation study on whether to use OMP module in Diffree's iterative results. Consecutive vanilla inpainting iterations (i.e., without OMP Module) lead to substantial image degradation.

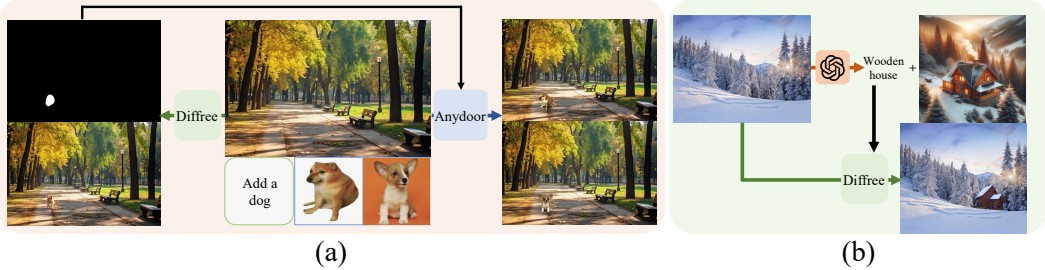

Figure 9: Applications combined with Diffree. (a): AnyDoor integrates Diffree's object position mask to add a specific object. (b): using GPT4V to plan what should be added.

## 4.4 APPLICATION

Diffree can be well combined with other methods for more expansion.

**With GPT4V** GPT4V (OpenAI, 2023) has a good ability to perceive and understand images, therefore we can use GPT4V for planning an object suitable for the image scene, seeing Figure 9. However, when tasked with adding the corresponding object without altering the background, DALL-E-3 (Betker et al., 2023) in GPT4, falls short.

**Post-Processing** AnyDoor (Chen et al., 2023) can insert a specific object into a designated area using a mask and object image. As depicted in Figure 9, Diffree provides a reasonable object mask to AnyDoor, facilitating the specific addition. DIffree also can effectively leverage the continuous progress in the mask-guided inpainting to generate superior images, as demonstrated in Table 1.

**Iterative Operation** In Figures 1 and 8, we present the results of iterative inpainting. Leveraging the predicted mask from the OMP module, Diffree can preserve the image background from cumulative degradation during successive inpainting. This holds potential applications within architectural and interior design domains. *See more applications in Section A6 of the Appendix.*

## 5 CONCLUSION

We propose a novel method, Diffree, that leverages a diffusion model with an object mask predictor for text-guided object addition. Beyond the method, we build a high-quality synthetic dataset, OABench, through a novel data collection method for this task. Diffree distinguishes itself by preserving background consistency without requiring additional masks, which solves shortcomings of previous text-guided and mask-guided object addition methods. The quantitative and qualitative results demonstrate the superiority of our method.

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

In this appendix, we present comprehensive elucidations as follows:

- Section A1: Additional results from our Diffree.
- Section A2: Visualizations of OMP-generated masks at various steps of Diffree's inference.
- Section A3: Detailed evaluation information of the reasonableness of object locations.
- Section A4: Analysis of the limitations of Diffree, including illustrative failure cases.
- Section A5: Data processing details for OABench.
- Section A6: Additional potential applications of Diffree.
- Section A7: Discussion of imprecise mask-guided methods and qualitative comparisons.
- Section A8: In-depth discussion of OMP module.
- Section A9: User study on overall satisfaction.
- Section A10: Generalization analysis of Diffree model.

## A1 MORE RESULTS

In Figure A1, we present the complete iterative process of Figure 1. Due to the ability to mix inpainted results with previous images using OMP masks, Diffree can perform multiple iterations. Additionally, we provide further visualization results of Diffree in Figures A2 and A3.

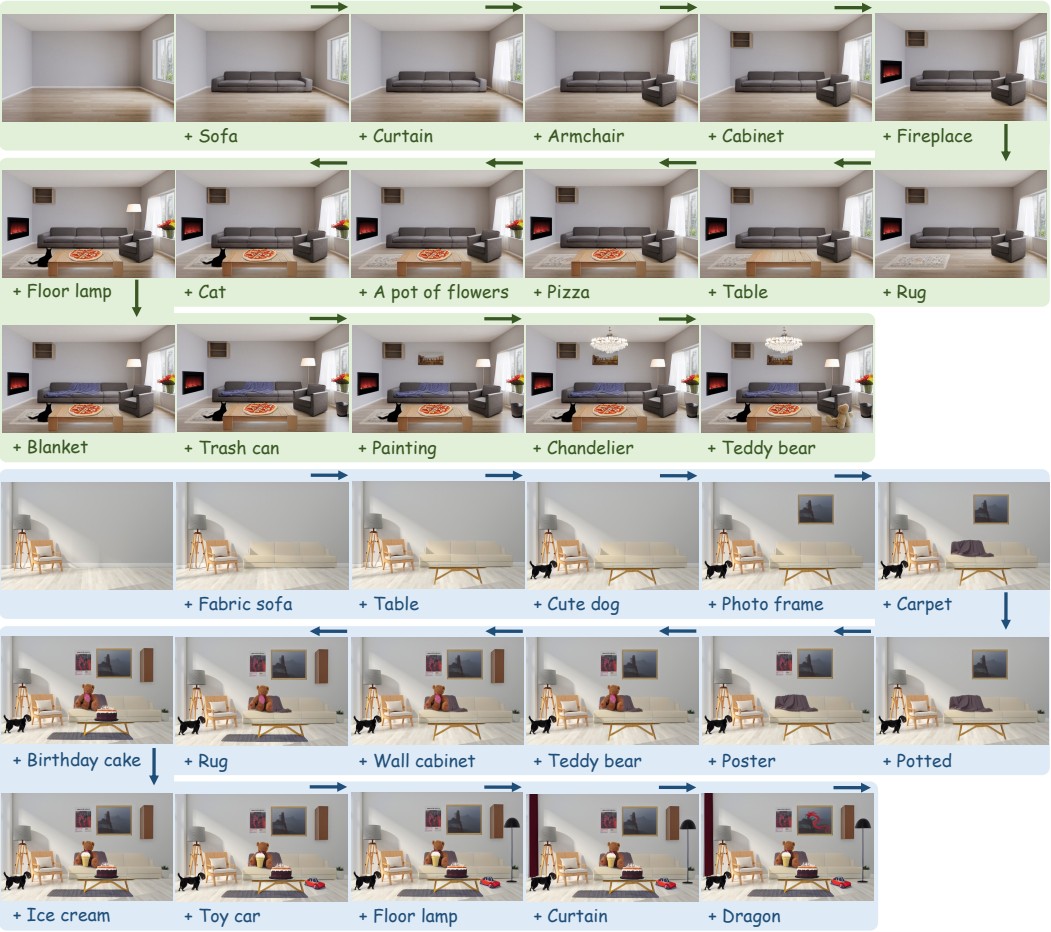

Figure A1: The complete iterative process of Diffree yields inpainted outcomes. The objects from text-guided are reasonably added in images while ensuring background consistency.

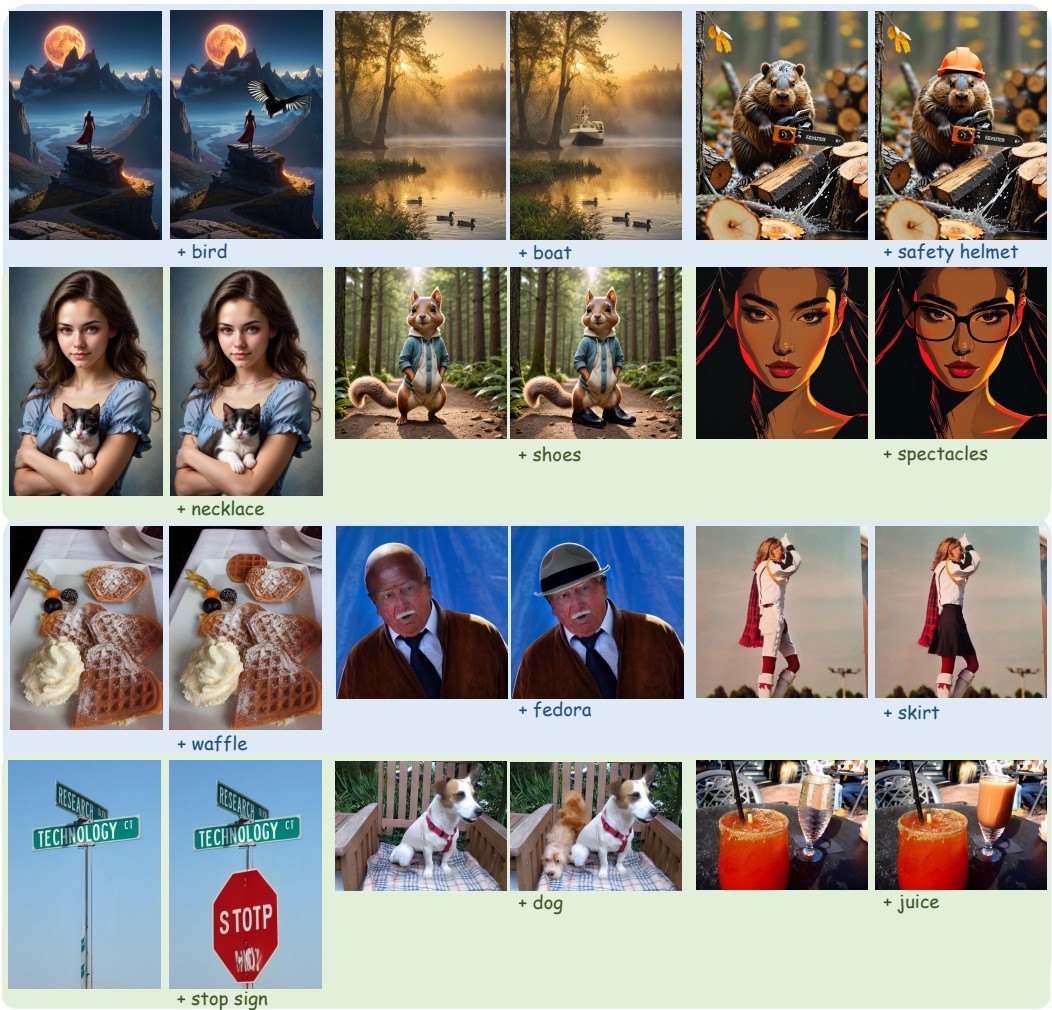

Figure A2: More visualization results of Diffree. In each pair, the image on the left is the input image, and the image on the right is generated by diffree.

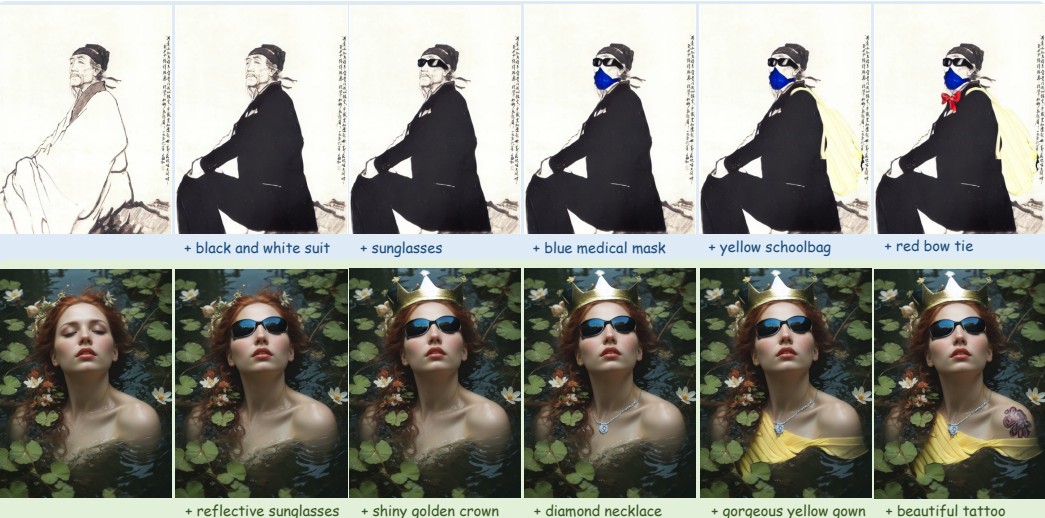

Figure A3: Diffree iteratively generates outputs that closely adhere to objects' descriptive attributes.

## A2   Mask at different steps

The OMP module predicts the mask via the reverse diffusion process, enabling early-stage mask prediction, as demonstrated in Figure A4. This significantly reduces computational time when integrating the mask generated by Diffree with other models (e.g., combined with AnyDoor (Chen et al., 2023) as illustrated in Figure 9).

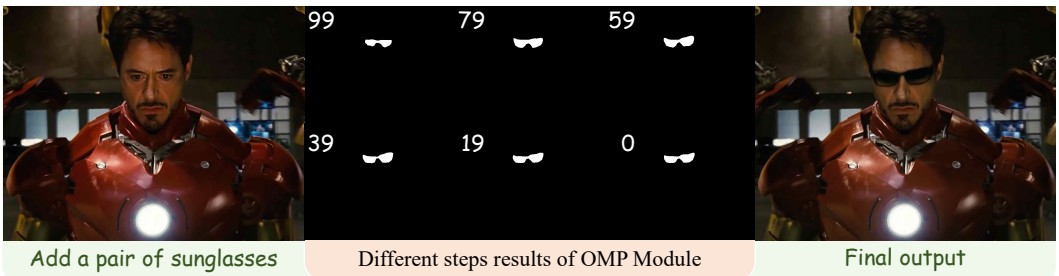

Figure A4: Visualization of masks from OMP at different steps of the diffusion inference process (totaling 100 steps) reveals that the mask of added objects can be initially obtained, such as during the first denoising step (step 99 out of 100).

## A3   Object Location evaluation

We use GPT4V to assess the reasonableness of the object's location. For each item evaluation, we provide input images and model-generated images, as well as a required caption and an evaluation instruction for GPT4V. The output comprises a dictionary a dictionary that includes assessment scores and rationale as demonstrated in Figure A5.

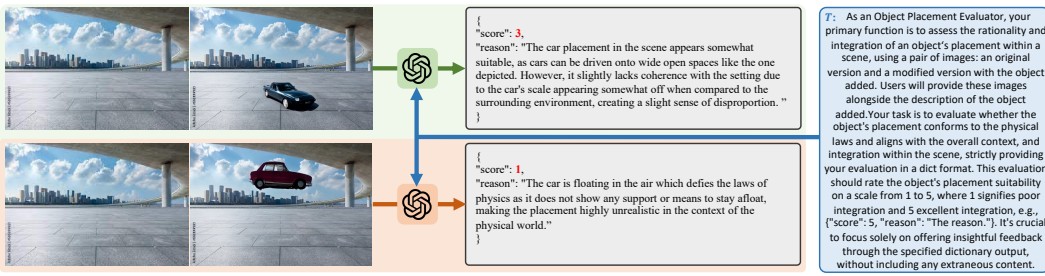

Figure A5: GPT4V shows good distinguishability in the reasonableness between objects.

We further conducted a user study to verify this ability. A random selection of 100 successful cases from InstructPix2Pix (Brooks et al., 2023) is compared with our outcomes. The comparison in Figure A6 demonstrates our significant advantage, with the win rate defined as the ratio of our wins to the total wins by either our method or InstructPix2pix. We observe that Diffree exhibits advantages in the reasonableness of the object's location, akin to the results presented in Table 1. It is worth noting that we only calculated the successful cases of InstructPix2pix, which only account for a small part of the complete results for InstructPix2pix.

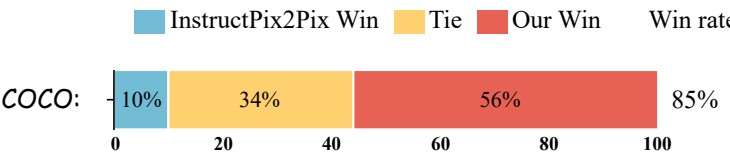

Figure A6: User study of location reasonableness on COCO dataset.

## A4  LIMITATION DISCUSSION

While Diffree has demonstrated remarkable performance across various metrics, several limitations remain. *Firstly*, the quality of our model is constrained by the visual fidelity of the inpainted dataset and thus by the inpainting model used for data generation. For instance, when specific objects both exist and require inpainting, our model occasionally exhibits a replacement phenomenon, as depicted in Figure A7. This is due to the presence of inferior partial data containing new objects after the inpainting stages, using the existing image inpainting model, of our data processing process. *Secondly*, this study primarily focuses on shape-free object inpainting (requiring only text), implying that users have no control over the shape. In future work, we aim to improve data quality and integrate benefits from mask-based methods to provide a transition from both.

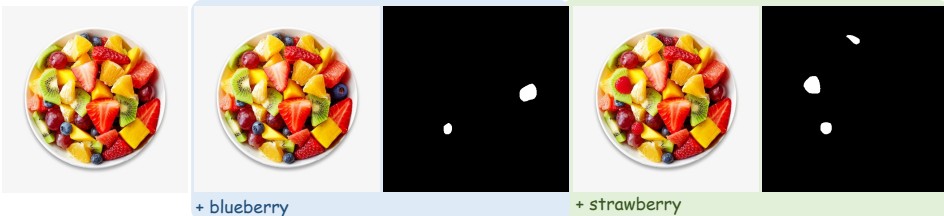

Figure A7: Failure cases. When the anticipated object is already present in the image, Diffree may occasionally fail to add the new object and instead replace the existing one.

## A5  DATA PROCESSING DETAILS OF OABENCH

This section delineates the data processing details for generating the synthetic dataset termed Object Addition Benchmark (OABench), comprising 74K real-world tuples, each containing an original image, an inpainted image, an object mask, and an object description.

### A5.1  COLLECTION AND FILTERING

We gather and refine instances suitable for image inpainting by applying a set of rules from the LVIS dataset (Gupta et al., 2019), a large and diverse instance segmentation dataset annotated for COCO (Lin et al., 2014) dataset. For an instance segmentation data item (i.e., one image containing multiple instances), we apply the following criteria to filter the appropriate cases:

Table A1: List of instance categories to be filtered out in our data processing.

| Category | Subcategory and Items (instance categories) |
|---|---|
| **Clothing and Accessories** | **Tops**: polo_shirt, sweatshirt, tank_top_(clothing), shirt, blouse, turtleneck_(clothing), cardigan, blazer, jacket, sweater, dress_hat, nightshirt; **Bottoms**: short_pants, skirt, trousers, jean; **Outerwear**: coat, parka, trench_coat, ski_parka, wet_suit; **Underwear and Sleepwear**: underwear, brassiere, nightshirt; **Jewelry**: anklet, necklace, bracelet, ring, broach, choker, barrette; **Belts and Ties**: belt_buckle, necktie, bolo_tie, suspenders; **Headwear**: bandanna, turban, veil; **Footwear**: shoe, boot, arctic_(type_of_shoe); **Other Accessories**: bolo_tie, tassel, wig; **Other Clothing Items**: breechcloth, dress, bridal_gown, ballet_skirt, vest, dress_hat. |
| **Food and Beverages** | **Beverages**: fruit_juice, cider, cocoa_(beverage), orange_juice, root_beer, lemonade, martini, cappuccino; **Sweets and Snacks**: brownie, lollipop, bubble_gum, jelly_bean, truffle_(chocolate), chocolate_mousse; **Ingredients and Condiments**: hummus, beef_(food), crabmeat, egg_yolk, salsa, cayenne_(spice), peanut_butter, crouton, string_cheese, broccoli, sausage, batter_(food), pea_(food), pepper, legume, hot_sauce, Tabasco_sauce, jam; **Main Dishes and Sides**: stew, lasagna, coleslaw, grits, mashed_potato, steak_(food), applesauce. |
| **Household Items** | **Kitchenware**: plate, paper_plate, drawer, garbage; **Cleaning Supplies**: cleansing_agent, gargle; **Linens and Textiles**: blanket, bath_mat, tablecloth, bath_towel, paper_towel, towel, bedspread; **Paper Products**: tissue_paper, napkin, envelope, plastic_bag, tape_(sticky_cloth_or_paper), toilet_tissue; **Packaging and Stationery**: envelope, tape_(sticky_cloth_or_paper), plastic_bag, tissue_paper, tinfoil; **Other Household Items**: mirror, place_mat, tarp, pacifier, bandage, surfboard, drumstick, mound_(baseball), wet_suit. |

**Category filtering** Initially, we manually annotate a list of categories which are typically considered parts of complete instances, as detailed in Table A1. We then remove the data from these categories, as they are typically challenging to remove from images and tend to produce unnatural inpainting results. Surprisingly, even after excluding these categories from the dataset, our model retains the capability to add such instances (e.g., shirts) to images.

**Size limitation** We limit the size of the instance mask at the pixel level through a maximum/minimum size ratio. Instances below the minimum size threshold are still highly probable to represent segments of complete instances. At the same time, instances that exceeded the maximum size threshold are very likely to be background (e.g., sky or grass). In our case, we set the maximum size ratio to 0.95 and the minimum size ratio to 0.01.

**Non-edge contact** Considering that instances touching the image boundaries are predisposed to incompleteness and pose challenges in background reconstruction, we exclude this subset of the data. We directly filter based on whether the mask information exists at the image edge.

**Integrity detection** To ensure the completeness of instance masks, we initially applied dilation and erosion operations to refine slightly separated mask segments. Subsequently, using OpenCV's connectedComponentsWithStats (Itseez, 2015), we analyze and sort the connected regions by size. If the ratio of the largest region's area to any other region's area exceeds a predefined integrity ratio threshold (set at 18), this largest region is considered representative of the instance.

**Non-hollow detection** We employ contour analysis to identify any potential hollow structures within the mask to ensure the non-occlusion of instance masks. Utilizing OpenCV's findContours function, we extract both the external and internal contours and designate instances with masks containing child contours as hollow. The hollow instances are considered to be partially obscured, while non-hollow masks are retained for further instance filtering.

**Aspect ratio filtering** We exclude instances with extreme aspect ratios, defined as those exceeding a threshold of 10 in either the horizontal or vertical dimension, due to their propensity to pose challenges for inpainting and their higher likelihood of representing partial objects.

**Non-occlusion detection algorithm** To ensure that the instances in our dataset are free from occlusions, we implement an occlusion detection algorithm based on the spatial relationships between instance masks. For each pair of instances with overlapping bounding boxes, determined by an Intersection over Union (IoU) exceeding a predefined threshold (set at 0.05), we compute the proportion of the overlapping area covered by each instance's mask. Specifically, we calculate the intersection area of their bounding boxes and assess how much of this area is occupied by each mask.

If the maximum of these coverage ratios is below an occlusion threshold (set at 0.15), the instances are considered non-occluding and are retained for further processing. However, if the minimum coverage ratio exceeds an area ratio threshold (set at 0.45), indicating significant mutual occlusion, both instances are discarded to prevent incomplete objects. In cases where one instance significantly occludes the other (evidenced by a larger coverage ratio), we discard the instance with the smaller coverage ratio. This selective filtering ensures that only non-occluded, fully visible instances are included in the dataset, enhancing the quality and reliability of the OABench.

### A5.2 DATA SYNTHESIS

We subsequently utilize the advanced mask-guided image inpainting method, PowerPaint (Zhuang et al., 2024), to remove targeted instances filtered in the prior phase. Before performing inpainting, we apply a dilation operation to the instance masks, as overly precise masks are insufficient for effective inpainting, using a slightly enlarged mask typically results in superior inpainting outcomes.

### A5.3 POST-PROCESSING

In the post-processing stage, our objective is to systematically filter out inpainting results that exhibit suboptimal object removal performance during the data synthesis phase. We then calculate the clip score using the instance name and the corresponding region of the inpainted image, where higher scores denote suboptimal removal efficacy. A preset threshold of 0.65 filters out instances with higher scores, thereby optimizing data quality and maintaining a reasonable data volume.

## A6 MORE APPLICATIONS

Recently, the popularity of image-to-video tools (e.g., Kling (Kuaishou AI, 2024)) has surged, allowing users to generate videos from single images. Due to Diffree's ability to reasonably add objects while maintaining the consistency of the image, the inpainted images remain suitable for image-to-video generation. Consequently, utilizing inpainted images from Diffree can enhance the flexibility of the generated video content and expand editing possibilities, as illustrated in Figure A8.

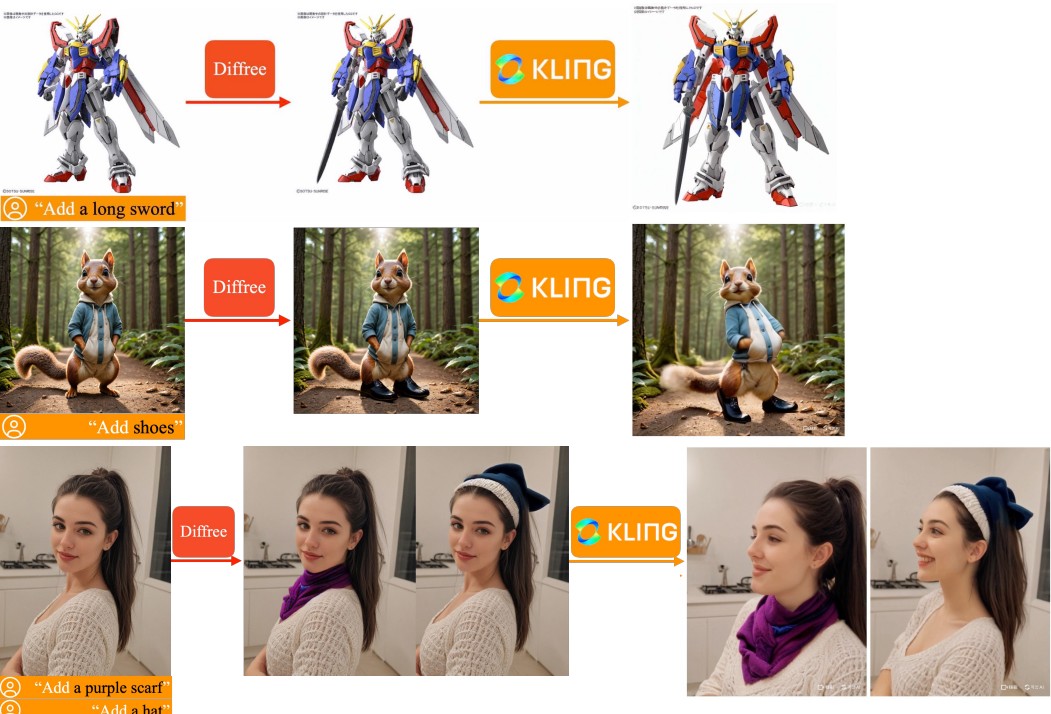

Figure A8: Using Kling video model to create videos from images inpainted with Diffree. The added objects seamlessly integrate into the generated video content.

## A7 DISCUSSION OF IMPRECISE MASK-GUIDED METHODS

Some mask-guided approaches do not require an exact mask condition, opting instead for imprecise masks (e.g., Glide (Nichol et al., 2022)) or bounding boxes (e.g., GLIGen (Li et al., 2023)).

These mask-guided methods only relax constraints on particular shapes, the necessity to specify reasonable size and position also requires inherent challenges. For instance, the results generated by GLIGen using masks that are not the appropriate size (either too large or too small) are clearly unexpected, as provided in Figure A9.

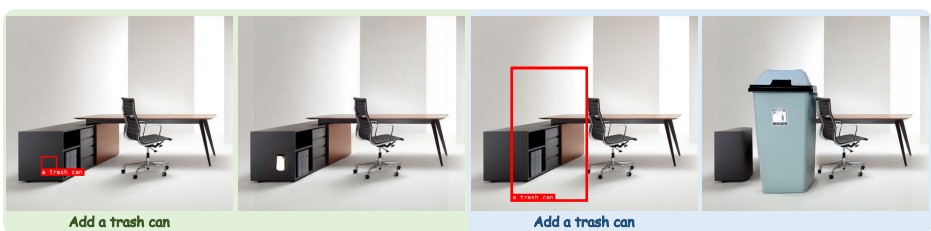

Figure A9: GLIGen's inpainting results under bounding box conditions with unexpected size. GLIGen still necessitates a predefined bounding box with reasonable size and position for inpainting.

Due to the additional mask condition of mask-guided methods compared to text-guided methods, we conduct a fair comparison of these methods by providing a complete mask or bounding box for methods necessitating additional conditions. As shown in Figure A10, although some mask-guided methods do not require precise mask information, they all fail without providing mask information. At the same time, InstructPix2pix exhibits a low success rate in adding an object while maintaining an unchanged background, as detailed in Figure 7.

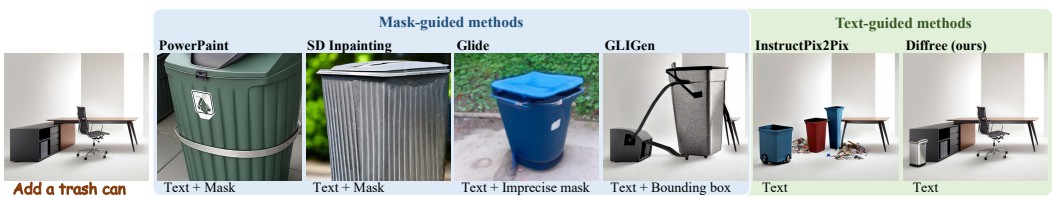

Figure A10: Qualitative comparison with other methods. The bottom represents the additional conditions required for each method. We provide comprehensive masks or bounding boxes for methods requiring them, thereby ensuring a fair and consistent comparison. All mask-guided methods fail in adding new objects only through text.

## A8 IN-DEPTH DISCUSSION OF OMP MODULE

In this section, we provide an in-depth discussion summarizing its role:

**1. The OMP module ensures background consistency, which is crucial for iterative additions.**

In mask-guided methods (e.g., PowerPaint Zhuang et al. (2024)), the instance mask is a required input. Typically, a post-processing step involves mix the synthesized object into the input image's background using the instance mask, ensuring the background remains unchanged.

In contrast, text-guided methods (e.g., InstructPix2pix Brooks et al. (2023) and our Diffree) do not require an instance mask as input, making this mix operation unavailable. This limitation can lead to quality degradation, especially in iterative additions. To the best of our knowledge, we are the first to introduce an output mask through the OMP module in text-guided shape free methods. This enables the mix operation and allows for iterative additions. We provided an ablation study (i.e., Figure 8) to evaluate the impact of omitting the OMP module on Diffree's iterative results. Without the OMP module, the background deteriorates rapidly after multiple steps, rendering further additions infeasible.

**2. The instance mask generated by the OMP module can be integrated with various existing works to develop exciting applications.**

As explained in the Appendix A6, the instance mask can be used in many applications that require a mask as input. For example, when combined with AnyDoor Chen et al. (2023), Diffree can achieve image-prompted object addition. We highlight a few additional points:

(1) Combining with shadow generation methods to produce realistic shadows

The task, shadow generation, aims to create plausible shadows for a composite foreground, given a composite image without foreground shadows and the foreground object mask Liu et al. (2024). However, existing mask-guided and text-guided inpainting methods pose challenges for shadow generation, especially for objects casting long shadows. These challenges arise for two reasons: first, there is a misalignment between the data masks and the actual shadows of objects (e.g., long shadows); second, for mask-guided method, it is difficult for users to draw the estimated masks of long shadow area with objects , in addition it is challenging for the model understands the respective parts of shadows and objects in the mask. Therefore, combining inpainting works with shadow generation works can lead to better results. In mask-guided methods, the shadow generation input can be derived from the input mask and output image. Thanks the OMP module's output mask, Diffree, as a text-guided method, effectively integrates with such methods to generate coherent objects and realistic shadows, as depicted in Figure A11.

(2) Providing a starting point for user or designer adjustments

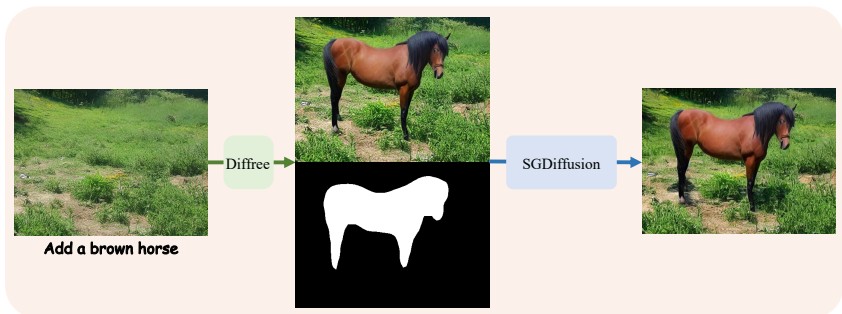

Figure A11: Diffree integrates with shadow generation work SGDiffusion Liu et al. (2024) to generate coherent objects and realistic shadows

In standard image processing, users or designers often need to make adjustments to achieve desired results. Diffree's output mask serves as a good starting point, making it easier for designers to refine the outcome. We emphasize that the OMP's mask output can be combined with evolving mask-based methods, serving as input for better results or continuous adjustments.

**3. OMP module outputs masks during the initial decoding steps, rather than after generation, under the proposed training process.**

OMP module's input is a concatenation of the latent representation of the input image and the expected output image's latent representation at each step, to output the object's mask area. If we only use the image pairs and object masks from our dataset to train OMP, it would generate masks only after the diffusion model's denoising process is completed during inference (e.g., after 100 steps). Therefore, we synchronize the training inputs of OMP with the diffusion training.

OMP module computes the predicted noise-free latent $\tilde{o}_t$ using the output from the diffusion model:

$$\tilde{o}_t = \frac{\tilde{z}_t - \sqrt{1 - \bar{\alpha}_t}\epsilon_\theta(\tilde{z}_t, z, \mathrm{Enc}_{\mathrm{txt}}(d), t)}{\sqrt{\bar{\alpha}_t}},$$

This $\tilde{o}_t$ can be derived from the denoising process and is available at each step, enabling mask prediction in the initial steps (i.e., Appendix A4). This allows us to quickly obtain a reasonable mask of the added objects without waiting for complete generation, facilitating integration with various applications. Training the OMP solely on image pairs and object masks would limit mask generation to after full denoising at inference time. By aligning the OMP's training inputs with those of the diffusion model and detaching gradients of $\tilde{o}_t$, we ensure independent optimization without interference. The independence of their loss functions and input consistency make separate training theoretically equivalent to joint training.

## A9 USER STUDY ON OVERALL SATISFACTION

We further conduct a user study on the overall satisfaction with all results between InstructPix2Pix and our Diffree in 1000 cases of COCO and OpenImages. As shown in Figure A12, the comparison results demonstrate that Diffree significantly outperforms InstructPix2Pix in user satisfaction. We believe this comprehensive user study effectively showcases the advantages of our method.

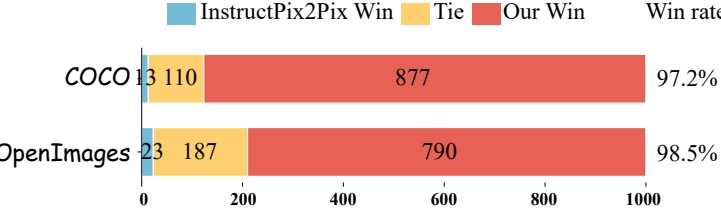

Figure A12: User study of overall satisfaction in 1000 cases of COCO Lin et al. (2014) and OpenImages Kuznetsova et al. (2020) dataset, with the win rate defined as the ratio of our Diffree wins to the total wins by either Diffree or InstructPix2pix.

## A10 GENERALIZATION ANALYSIS OF DIFFREE MODEL

### A10.1 DISCUSSION ON ARTIFACTS IN SYNTHETIC DATASETS

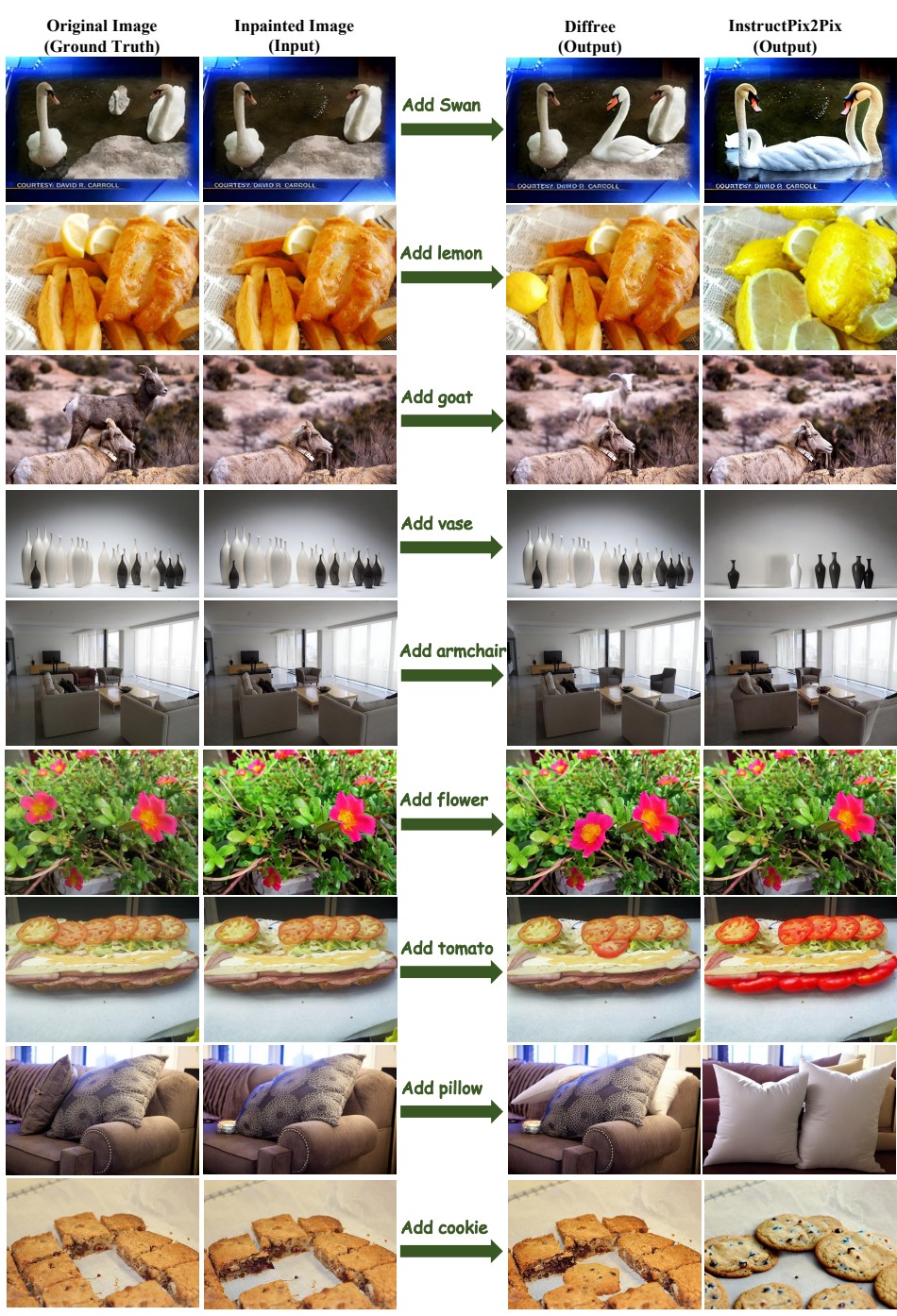

Figure A13: Diffree's results of objects were removed using an inpainting model (e.g., PowerPaint) from an image containing multiple identical objects and then added back using Diffree to see if it would be added onto the same position and size. The added objects appeared at different positions and sizes, indicating no overfitting to artifacts and demonstrating generalization.

we conduct additional experiments where objects were removed using an inpainting model (e.g., PowerPaint Zhuang et al. (2024)) from an image containing multiple identical objects and then added back using Diffree to see if it would be added onto the same position and size. As shown in Figure A13, the added objects appeared at different positions and sizes, indicating no overfitting to artifacts and demonstrating generalization.

### A10.2    DISCUSSION ON OUT-OF-DISTRIBUTION (OOD) OBJECTS OF OABENCH

Diffree demonstrates strong performance when handling out-of-distribution (OOD) objects not included in our OABench dataset. As detailed in Appendix A5.1, our model is capable of adding various objects absent from the training data. For example, Figures A1 to A3 illustrate that Diffree can successfully add objects like "dragon", "necklace" or other OOD items.

This generalization capability stems from our fine-tuning approach based on the pre-trained SD1.5, which inherently can generate various objects from text descriptions. Therefore, even objects not present in our dataset can be added by Diffree. This reflects Diffree's robustness and adaptability to different image styles and unseen objects, making it applicable to a wide range of scenarios.

### A10.3    DISCUSSION OF RESPONSES TO SPECIFIC OR INTERACTIVE PROMPTS

**1. specific object attributes**

Despite the generic labels during training, Diffree demonstrates strong generalization and effectively responds to detailed, fine-grained prompts. As shown in Figure A3, Diffree successfully follows instructions such as "add shiny golden crown" and "add reflective sunglasses," producing appropriate additions that match the detailed descriptions. This capability stems from our fine-tuning based on the pre-trained SD1.5, which inherently generates objects with specific attributes from text descriptions. This reflects Diffree's robustness and adaptability, making it applicable to more scenarios.

**2. context-based interactions**

To enhance precise control in object addition, we extended our model by re-labeling our dataset with accurate location descriptions using GPT-4o-mini OpenAI (2024) and retrained our model based on pre-trained SD1.5 with these detailed annotations and original annotations.

We provide the following inputs from our OABench to GPT-4o-mini for precise re-labeling: (1) Cropped area image of the object to help GPT-4o-mini understand the object's attributes. (2) Original image containing the object to establish context and correspondence. (3) object label description. (4) A Prompt of the object to guide the task: *"You will be provided with the following: 1.A real image of a scene. 2.A cropped image of a specific object from the scene. 3.The category text of the object (e.g., 'cup', 'chair'). Based on this information, generate a concise description of the object's appearance and its spatial position in the scene. The description should be no longer than 20 words and focus solely on the object and its immediate spatial relationship. Example: 'A transparent cup on the table.' Avoid adding unnecessary context or details beyond the object's appearance and position."* We followed the training process outlined in the Section 3, with the exception that we used both the original descriptions and the relabeled descriptions as text prompts for training.

We conducted experiments using contextually detailed prompts specifying different locations within the same scene or involving multiple similar objects. As shown in Figure A14, our model can accurately add objects based on context-related descriptions. This demonstrates that Diffree can be extended to handle more precise control. We plan to further enhance this capability by constructing larger datasets with precise annotations.

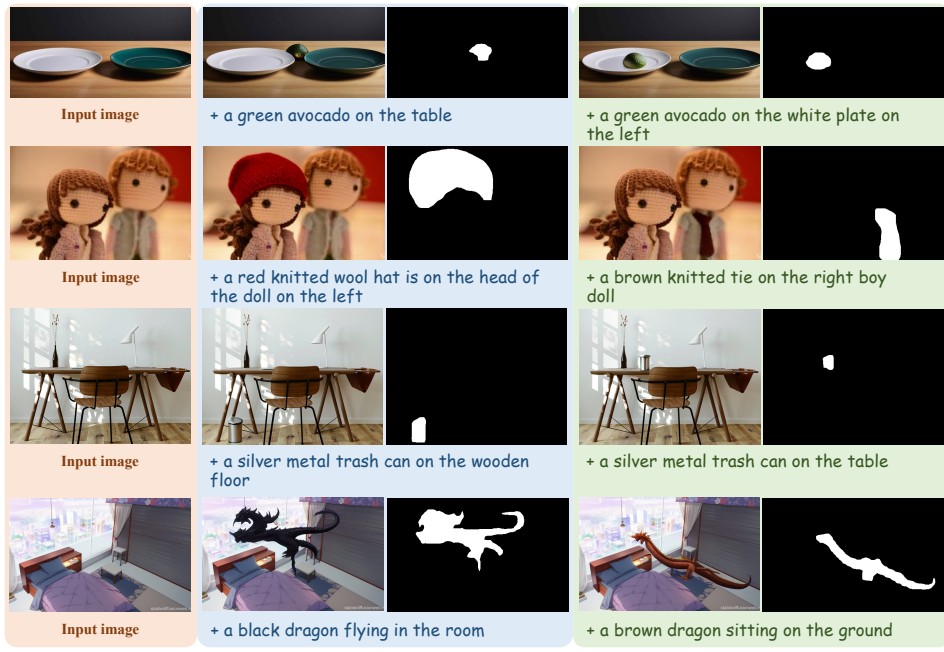

Figure A14: Diffree's results were obtained using contextually detailed prompts specifying different locations within the same scene or involving multiple similar objects. We extended our model by re-labeling our dataset with accurate location descriptions using GPT-4o-mini OpenAI (2024) and retrained Diffree with these detailed annotations and original annotations.

