# OpenReview forum: "Diffree: Text-Guided Shape Free Object Inpainting with Diffusion Model"
_ICLR.cc/2025/Conference — Submitted to ICLR 2025_

### Official Review · Reviewer_js6X · 2024-10-26

**Soundness:** 2
**Presentation:** 2
**Contribution:** 2
**Rating:** 5
**Confidence:** 4

**Summary:**

This paper proposes an object addition method called Diffree. Diffree comprises an object mask prediction (OMP) module, which could be used to generate masks through textual inputs. Thus Diffree can be regarded as a mask-free inpainting model. The authors also provide a synthetic dataset called OABench with 74k real-world object removal tuples built by PowerPaint. Extensive experiments verify the effectiveness of Diffree.

**Strengths:**

1. This paper proposed a sophisticated and reliable data collection process to achieve high-quality image removal tuples, which would potentially strengthen many other inpainting-based tasks.

2. The authors show some promising and impressive object addition results in the appendix, including both real-world and AI-generated images.

**Weaknesses:**

1. As mentioned in the main paper, Diffree is trained to predict precise object masks, which is not convincing for both object removal and object addition. Because most objects contain shadows or reflections, which are not included in object masks. Without proper shadow and reflection generation, the inserted objects suffer from "copy and paste" artifacts. For example, no shadows are generated in the "Coconut palm" of Figure 3, and no reflections are generated in the "boat" of Figure A2. The metric of background consistency is also problematic. I think this is the main drawback of this method.

2. Missing discussion and ablation about the necessity of the joint training of OMP and diffusion model. Why not independently train OMP to predict the mask.

3. The generalization of the Diffree should be concerned. As shown in Figure 9 (a), the generation failed while the mask was very small.

4. For the object addition guided completely by texts, the authors should provide more results to verify whether the proposed method could handle different orientational descriptions. For example, these prompts should be considered: "on left/right/up/bottom of...".

**Questions:**

1. Why not use Diffree to address mask-free removal? This should also be addressed by OABench.

2. What did "with/without mix with previous image" mean in Figure 8?

---

> ### Author Response · Authors · 2024-11-21
> **Response to Reviewer js6X (Part: 1/3)**
>
> We sincerely thank Reviewer js6X for their valuable time and insightful comments on our work. We have summarized the updating of our revision in Summary of Paper Updating and provided an in-depth discussion about our Object Mask Predictor (OMP) module in Genera Response. Below, we address each of your concerns individually.
>
> **Q1.1:**  Diffree is trained to predict precise object masks, most objects contain shadows or reflections, which are not included in object masks. Without proper shadow and reflection generation, the inserted objects suffer from "copy and paste" artifacts.
>
> **A1.1:**  Thank you for highlighting the importance of shadows and reflections in object insertion. We acknowledge that including these elements is crucial for realistic results. However, Diffree effectively handles them for the following reasons:
>
> 1. Diffree is trained to add objects in images through text and output the masks of the objects, instead of predict object masks. The model learns to render objects within the scene context, naturally incorporating appropriate shadows and reflections to generated images.
>
> 2. **Use of Real Images in Training:** Our data collection process utilizes real images with objects as training targets, unlike previous methods that may rely on synthetic data. This ensures that the generated objects in Diffree inherit the inherent shadows and reflections present in real-world scenarios, leading to seamless integration with the background (e.g., avocado in Figure A14).
>
> 3. **Objects Casting Long Shadows:** We recognize that existing mask-guided and text-guided inpainting methods pose challenges for shadow generation, especially for objects casting long shadows. These challenges arise for two reasons: first, there is a misalignment between the data masks and the objects and actual shadows (e.g., long shadows); second, for mask-guided method, it is challenging for users to draw the estimated masks of long shadow area with objects , which is also challenging for the model understands the respective parts of shadows and objects in the mask. Therefore, combining inpainting works with shadow generation works can lead to better results for objects casting long shadows.
>
>    To address this, Diffree’s OMP module generates object masks that can be integrated with shadow generation methods [1], as shown in Figure A11 of the Appendix. These methods are designed to produce plausible shadows for composite objects using the object mask and the composite image. By combining Diffree with such techniques, we can generate coherent objects with realistic long shadows, enhancing the overall visual quality.
>
>    This integration reflects the good scalability of Diffree and its ability to produce realistic results without “copy and paste” artifacts.
>
>
> [1] Liu, Qingyang, et al. "Shadow Generation for Composite Image Using Diffusion Model." *Proceedings of the IEEE/CVF Conference on Computer Vision and Pattern Recognition*. 2024.
>
> **Q1.2:**  The metric of background consistency is also problematic.
>
> **A1.2:**  Thank you for expressing concerns about the metric used for evaluating background consistency. We agree that an appropriate metric is essential for accurate assessment.
>
> We use the Learned Perceptual Image Patch Similarity (LPIPS) metric [1], which is widely recognized for measuring perceptual similarity between images. To specifically evaluate background consistency, we focus on the regions outside the generated object:
>
> 1. We replace the area of the generated object in the output image with the corresponding area from the input image. This isolates the background for comparison.
> 2. We compute the LPIPS score between the input image and this modified output image, effectively measuring differences only in the background regions.
>
> This approach allows us to quantify how closely the background in the generated image matches the original background. An ideal score would indicate no perceptual difference, signifying perfect background consistency.
>
> We believe this method provides a robust and accurate evaluation of background consistency, ensuring that any changes in the background introduced during object insertion are appropriately measured.
>
>
>
> [1] Zhang, Richard, et al. "The unreasonable effectiveness of deep features as a perceptual metric." CVPR 2018.
>
> **Q2:** Missing discussion and ablation about the necessity of the joint training of OMP and diffusion model.
>
> **A2:** We appreciate your suggestion to elaborate on the necessity of ablation study and joint training for the OMP module and the diffusion model. In Figure 8, we present an ablation study demonstrating the impact of omitting the OMP module. The results show that without the OMP module, background consistency deteriorates rapidly in iterative additions. OMP is a lightweight module that outputs the mask of added objects. Please refer General response (Point 3) for more details about the joint training of OMP and diffusion model.

---

> ### Author Response · Authors · 2024-11-21
> **Response to Reviewer js6X (Part: 2/3)**
>
> **Q3:** The generalization of the Diffree should be concerned. As shown in Figure 9 (a), the generation failed while the mask was very small.
>
> **A3:** Thank you for raising the concern about Diffree’s generalization, specifically regarding Figure 9(a). We would like to clarify that:
>
> **Figure 9(a) Clarification:** The image on the right side of Figure 9(a) is generated by AnyDoor [1] using Diffree's mask, not generated by Diffree. The dog generated by Diffree (left side) is depicted facing away, which we consider a plausible variation and maintains consistency with the background.
>
> The following is the evidence we believe supports generalization:
>
> 1. **Use of Real Images:** Diffree is trained using real images as outputs, avoiding artifacts commonly introduced by synthetic datasets or inpainting models. This ensures that the generated objects are naturally integrated with the background.
>
> 2. **Overfitting Analysis:** To assess potential overfitting in input, we further conduct experiments where we removed using an inpainting model (e.g., PowerPaint [2]) from an image containing multiple identical objects and then added back using Diffree to see if it would be added onto the same position and size. As shown in Figure A13 of the Appendix, the added objects appeared at different positions and sizes, indicating no overfitting to artifacts and demonstrating generalization.
>
> 3. **Diverse generation conditions:** We have successfully applied Diffree to:  1) Real images without artifacts. 2) Anime images (Figures A2 and A3 in the Appendix). 3) Adding objects not present in our dataset, demonstrating robustness to out-of-distribution scenarios.
>
> We acknowledge certain limitations of Diffree in Section A4 of the Appendix. However, the evidence suggests that Diffree generalizes well across different contexts, maintaining high-quality results without significant failure due to mask size or overfitting.
>
>
>
> [1] Chen, Xi, et al. "Anydoor: Zero-shot object-level image customization." CVPR 2024.
>
> [2] Zhuang, Junhao, et al. "A task is worth one word: Learning with task prompts for high-quality versatile image inpainting." ECCV 2024.
>
> **Q4:** For the object addition guided completely by texts, the authors should provide more results to verify whether the proposed method could handle different orientational descriptions.
>
> **A4:** Thank you for this important suggestion. To enhance precise control in object addition, we extended our model by re-labeling our dataset with accurate location descriptions using GPT-4o-mini and retrained Diffree with these detailed annotations and original annotations.
>
> To address your concerns, we conducted experiments using contextually detailed prompts specifying different locations within the same scene or involving multiple similar objects. As shown in Figure A14, our model can accurately add objects based on context-related descriptions (e.g., “Add a red knitted wool hat on the head of the doll on the left”). This demonstrates that Diffree can be extended to handle more precise control. We plan to further enhance this capability by constructing larger datasets with precise annotations.

---

> ### Author Response · Authors · 2024-11-21
> **Response to Reviewer js6X (Part: 3/3)**
>
> **Q5:** Why not use Diffree to address mask-free removal? This should also be addressed by OABench.
>
> **A5:** We appreciate your interest in extending Diffree to mask-free text-guided object removal. While OABench could indeed facilitate training for such tasks, our focus in this paper is on object addition, which presents unique and significant challenges:
>
> 1. **Complexity of Object Addition:** Adding new objects requires the model to determine appropriate placement, scale, and integration with the existing scene, without any prior information about the object’s presence.
> 2. **Existing Object Removal:** In object removal, the target object is already present, and its location and size are known, making the task more straightforward as the model can directly manipulate the existing content.
>
> We chose to concentrate on object addition to address these challenges comprehensively. Developing Diffree for mask-free object removal is an interesting direction for future work, and we believe our approach and dataset could be adapted to support this extension.
>
> **Q6:** What did "with/without mix with previous image" mean in Figure 8?
>
> **A6:** Thank you for pointing out the need for clarification. In Figure 8, “with/without mix with previous image” refers to whether we perform a mix operation using the mask generated by the OMP module during iterative object additions. (1) With Mix: We use the OMP generated mask to mix the newly generated object into the original input image, ensuring that the background remains consistent across iterations. (2) Without Mix: We directly use the output image from the previous iteration without mix, which can lead to background degradation over multiple additions.
>
> We have updated the caption of Figure 8 in the revised manuscript to make this clearer. The ablation study demonstrates that using the OMP module and performing the mix operation significantly improves background consistency and quality during iterative additions. This capability was not possible in previous text-guided methods due to the lack of available object masks.
>
> For more details about the OMP module and its role, please refer General Response.

---

> ### Author Response · Authors · 2024-11-24
> **Last three day reminder and looking forward to further discussion**
>
> Dear Reviewer js6X,
>
> Thanks again for your valuable time and insightful comments. We hope that our response can address your concerns. As the deadline for Author/Reviewer discussion period is approaching, we really appreciate if you can let us know whether there still exists any further question about the paper or the response. We are looking forward to further discussion.
>
> Best regards,
>
> Authors of Paper 2788

---

> > ### Comment · Reviewer_js6X · 2024-11-24
> > **Thanks for the rebuttal**
> >
> > Thanks for the rebuttal.
> > Some concerns are still retained.
> >
> > 1. Although the authors provide the pipeline of object addition and shadow generation, this solution is not a general and robust way to address various scenarios. The final performance would be highly influenced by the harmonization or shadow generation methods.
> >
> > 2. Thanks for the results and solution to handle object addition with different orientational descriptions. I think this property should be discussed more, and more experimental metrics should be included. Without the description control, mask-free object addition works as a random mask generation, failing to show the user's intention. And the advancement of mask-free object addition is not obvious compared to mask-controlled image inpainting, while the latter could be perfectly controlled for locations, shapes, and sizes.

---

> > > ### Author Response · Authors · 2024-11-24
> > > **Response to Reviewer js6X**
> > >
> > > Dear Reviewer js6X,
> > >
> > > Thank you for your continued engagement with our work and for sharing your valuable insights. We appreciate the opportunity to address your remaining concerns.
> > >
> > >
> > > **Regarding your first concern:**
> > >
> > > *“Although the authors provide the pipeline of object addition and shadow generation, this solution is not a general and robust way to address various scenarios. The final performance would be highly influenced by the harmonization or shadow generation methods.”*
> > >
> > > We would like to address this concern from the following perspectives:
> > >
> > > 1. **Our method performs well with object reflections and subtle shadows.** For example, in Figure A14 where an avocado is added, the lighting on the plate indicates that the light source comes from the upper right corner. Among multiple results, Diffree successfully renders the new avocado with natural lighting on its right side. Additionally, shadows are correctly displayed on the underside of the avocado.
> > > 2. **Generation objects casting long shadows is a challenge common to all inpainting methods.**
> > > 3. **Object shadow generation is an active research area in itself [1].**
> > > 4. **Our Diffree method achieves a 98% success rate in object addition, significantly improving over previous methods that had only a 17% success rate.**
> > >
> > > We hope that future work can address this issue more comprehensively. Currently, our method focuses on resolving the low success rate and poor consistency in text-guided object addition.
> > >
> > > [1] **Awesome-Object-Shadow-Generation**: A collection of research on object shadow generation. https://github.com/bcmi/Awesome-Object-Shadow-Generation
> > >
> > > **Regarding your second concern:**
> > >
> > > *“Thanks for the results and solution to handle object addition with different orientational descriptions. I think this property should be discussed more, and more experimental metrics should be included. Without the description control, mask-free object addition works as a random mask generation, failing to show the user's intention. And the advancement of mask-free object addition is not obvious compared to mask-controlled image inpainting, while the latter could be perfectly controlled for locations, shapes, and sizes.”*
> > >
> > > We would like to address this concern from the following perspectives:
> > >
> > > 1. **Text-guided object addition indeed presents significant challenges compared to mask-guided inpainting.** Without explicit masks, the model must infer the object’s location, size, and orientation solely from textual descriptions and the image context. This requires a higher level of scene understanding and contextual reasoning.
> > >
> > > 2. **Text-guided methods that do not include context are more challenging than those that include context.** For example, the prompts “Add a cup” and “Add a cup on the table” may result in similar images, but the former relies entirely on the model’s reasoning to determine placement, whereas the latter provides a location hint. This highlights the increased difficulty when the model must infer more information from less guidance.
> > >
> > >    This is also where we believe the difficulty of object addition lies compared to object editing/removing: the hint already present in the image.
> > >
> > > 3. **Mask-guided and text-guided methods are two compatible parallel research fields, each with its own advantages [1].** Mask-guided methods offer precise control over object placement, size, and shape but require users to provide accurate masks. Text-guided methods, on the other hand, are more accessible to users who may not have the skills or tools to create precise masks.
> > >
> > > 4. **The practical application of text-guided methods demonstrates their importance.** Commercial tools utilizing text-guided methods exist [2] recently, indicating the relevance and demand for such approaches.
> > >
> > > As mentioned in point 2, our main goal was to allow the model to infer object positions, sizes, and shapes autonomously. While adding detailed location descriptions can improve control, it also reduces the model’s need to reason about the scene. We believe that advancing mask-free object addition is important because it pushes the boundaries of AI-driven image editing, making it more accessible and user-friendly.
> > >
> > > [1] Huang, Yi, et al. "Diffusion model-based image editing: A survey." *arXiv preprint arXiv:2402.17525* (2024).
> > >
> > > [2] https://team.doubao.com/en/special/seededit
> > >
> > >
> > >
> > > Thank you again for your thoughtful feedback. Your insights help us to refine our work and highlight areas for further improvement. We are dedicated to advancing this research and are optimistic that addressing these challenges will contribute significantly to the field.
> > >
> > > Please feel free to share any additional thoughts or questions you may have.
> > >
> > >
> > >
> > > Best regards,
> > >
> > > Authors of Paper 2788

---

### Official Review · Reviewer_zVNG · 2024-10-30

**Soundness:** 2
**Presentation:** 1
**Contribution:** 2
**Rating:** 3
**Confidence:** 3

**Summary:**

The paper introduces Diffree, a text-to-image (T2I) model aimed at text-guided object addition within images, a process where new objects are integrated into a scene while maintaining visual consistency in lighting, texture, and spatial orientation. Unlike previous approaches, Diffree avoids the need for manually defined masks, instead using a synthetic dataset (OABench) for training. OABench, composed of image pairs with and without specified objects, is created by advanced image inpainting techniques. Trained on OABench, Diffree incorporates a Stable Diffusion backbone with an Object Mask Predictor (OMP) module to predict object placement. Results indicate that Diffree achieves high success in preserving background coherence, achieving spatially plausible integration, and producing visually realistic objects.

**Strengths:**

- **Originality**: Diffree’s approach of shape-free object addition guided solely by text is novel, significantly enhancing usability by eliminating the need for manual mask definitions. This innovation in user experience represents a unique contribution.
- **Clarity**: Overall, the dataset creation process, and evaluation metrics are well-described, with figures that aid understanding of Diffree’s operational and comparative performance.

**Weaknesses:**

- **Minor Typographical Issue**: There is a missing period at the end of line 101, which should be corrected for clarity.
- **Dataset Limitation in Prompt Detail**: Since the dataset primarily relies on the COCO dataset, prompts are often generic object labels rather than detailed, fine-grained descriptions. This limitation can hinder the model's ability to respond to nuanced or interactive prompts, such as requests for specific object attributes or context-based interactions.
- **Methodology Clarity**: The description of the proposed methodology, especially the Object Mask Predictor (OMP) module, could benefit from further clarification. For example, more detail on why the image latent and latent noise are concatenated as inputs for the OMP module during both training and inference phases would help improve comprehension.
- **Figure 6 Presentation**: Figure 6 needs a more detailed caption to allow readers to follow the workflow easily by reading only the figure caption. The current figure layout lacks clarity, making it challenging for readers to understand the process.
- **Inconsistency in Visuals (Figure 1)**: In Figure 1, example 2 appears visually inconsistent, with an unrealistic rendering of the added object. This raises questions about the model's object realism under certain conditions.
- **Lack of User Study**: Including a user study would strengthen the evaluation of the model, particularly to capture subjective aspects of user satisfaction and perceived realism in object integration.
- **Limited Comparative Evaluation**: The model is only compared against PowerPaint and InstructPix2Pix. Including additional comparison baselines, such as recent text-guided inpainting methods or general object addition approaches, would provide a more comprehensive assessment.
- **Ambiguity in Object Placement in Complex Scenes**: The method struggles with complex images involving multiple entities (e.g., two people). When required to add an object to one specific entity, the model may face ambiguity, as there is limited functionality for users to adjust the mask and specify the target placement.

**Questions:**

1. How does Diffree handle ambiguities in scenes with multiple entities (e.g., if there are two people in an image, how does it decide which one to add the object to)? Could user input be incorporated to clarify target placement in such cases?
2. Could more fine-grained prompts be incorporated or generated to enhance the model's ability to handle specific requests for object details and interactions within a scene?

---

> ### Author Response · Authors · 2024-11-21
> **Response to Reviewer zVNG (Part: 1/3)**
>
> We would like to express our sincere gratitude to Reviewer zVNG for their valuable time and effort in reviewing our work. We have summarized the updating of our revision in Summary of Paper Updating and provided an in-depth discussion about our Object Mask Predictor (OMP) module in Genera Response. Below, we address each identified weakness.
>
> **Q1:** Minor Typographical Issue: a missing period at the end of line 101.
>
> **A1:**  Thank you for pointing this out. We have corrected the typo in the updated manuscript.
>
> **Q2:** Since the dataset primarily relies on the COCO dataset, prompts are often generic object labels rather than detailed, fine-grained descriptions. This limitation can hinder the model's ability to respond to specific or interactive prompts.
>
> **A2:**  We appreciate your concern regarding the use of generic object labels in training. We address your points on specific object attributes and context-based interactions separately:
>
> 1. **specific object attributes**
>
> Despite the generic labels during training, Diffree demonstrates strong generalization and effectively responds to detailed, fine-grained prompts. As shown in Figure A3 of the Appendix, Diffree successfully follows instructions such as “add shiny golden crown” and “add reflective sunglasses,” producing appropriate additions that match the detailed descriptions. This capability stems from our fine-tuning approach based on the pre-trained SD1.5, which inherently generates objects with specific attributes from text descriptions. This reflects Diffree’s robustness and adaptability, making it applicable to a wide range of scenarios.
>
> 2. **context-based interactions**
>
> To enhance precise control in object addition, we extended our model by re-labeling our dataset with accurate location descriptions using GPT-4o-mini and retrained Diffree with these detailed annotations and original annotations.
>
> To address your concerns, we conducted experiments using contextually detailed prompts specifying different locations within the same scene or involving multiple similar objects. As shown in Figure A14, our model can accurately add objects based on context-related descriptions (e.g., “Add a red knitted wool hat on the head of the doll on the left”). This demonstrates that Diffree can be extended to handle more precise control. We plan to further enhance this capability by constructing larger datasets with precise annotations.
>
> **Q3:** Further clarification about the description of Object Mask Predictor (OMP) module. why the image latent and latent noise are concatenated as inputs for the OMP module during both training and inference phases?
>
> **A3:** Thank you for seeking clarification on the OMP module. We further discusse the OMP module in detail in Appendix A7 and the General Response (Point 3). The concatenation of the image latent and noise-free latent allows the OMP module to access both the input image information and the predicted latent representation. This combination enables the OMP to predict the object mask during the initial decoding steps, as the noise-free latent contains essential information about the generated object. By utilizing both inputs, the OMP module can produce accurate masks corresponding to the objects being added, facilitating efficient integration with various applications. We also updated the caption of Figure 6 to provide a clearer explanation of our Diffree's training process.
>
> **Q4:** Figure 6 Presentation: Figure 6 needs a more detailed caption to allow readers to follow the workflow easily by reading only the figure caption.
>
> **A4:** Thank you for the suggestion. We have updated the caption of Figure 6 to provide a clearer explanation of our Diffree's training process.

---

> ### Author Response · Authors · 2024-11-21
> **Response to Reviewer zVNG (Part: 2/3)**
>
> **Q5:** Inconsistency in Visuals (Figure 1): In Figure 1, example 2 appears visually inconsistent, with an unrealistic rendering of the added object. This raises questions about the model's object realism under certain conditions.
>
> **A5:** We appreciate your observation regarding the visual consistency in Figure 1. In this example, we performed 16 iterations of object additions to the image, with many newly added objects interacting with previously added ones (e.g., [fabric sofa, table, carpet, teddy bear, ice cream, birthday cake]). Achieving such a high number of additions with complex interactions is challenging and was nearly impossible with previous text-guided methods.
>
> We recognize that multiple overlapping additions can degrade realism due to compounded inpainting effects on image quality in these overlapping areas, challenging all methods. Conversely, this scenario highlights the importance of our OMP module in maintaining background consistency and overall image quality. As shown in Figure 8, without the OMP module, the image quality deteriorates significantly after multiple additions.
>
> Despite these challenges, the positions and sizes of the new objects remain reasonable and the perspective relationship and reflection of objects are consistent with the environment, demonstrating Diffree’s capability in rational object addition even in complex scenarios.
>
> **Q6:** Lack of User Study, particularly on user satisfaction and perceived realism
>
> **A6:** Thank you for pointing out the importance of user studies in evaluating satisfaction and perceived realism. The user study results has provided in Figure A6 of the Appendix on the reasonableness of added object locations.
>
> For better address your concerns, we conduct a further user study to assess overall satisfaction. We compare the results of Diffree and InstructPix2Pix across 1,000 cases from the COCO and OpenImages datasets. Participants were asked to evaluate which method produced more satisfactory results. The results are as follows:
>
> - User study of overall satisfaction in 1000 cases of COCO and OpenImages dataset, with the win rate defined as the ratio of our Diffree wins to the total wins by either Diffree or InstructPix2pix.
>
>   | Dataset        | InstructPix2Pix Win | Tie  | Diffree Win | Win Rate |
>   | -------------- | ------------------- | ---- | ----------- | -------- |
>   | **COCO**       | 13                  | 110  | 877         | 98.5%    |
>   | **OpenImages** | 23                  | 187  | 790         | 97.2%    |
>
> These results demonstrate that Diffree significantly outperforms InstructPix2Pix in terms of user satisfaction and perceived realism.  We believe this comprehensive user study effectively showcases the advantages of our method.
>
> **Q7:** The model is only compared against PowerPaint and InstructPix2Pix. Including additional comparison baselines would provide a more comprehensive assessment.
>
> **A7:** Thank you for your question. We have included qualitative comparisons with additional methods [1,2,3,4] in Figure A10 of the Appendix. We would like to emphasize two key points:
>
> 1.**Different Data Collection Processes:** Our data collection process differs significantly from other text-guided methods. Previous methods often struggle with object addition, yielding low success rates (e.g., 18% for InstructPix2Pix, as shown in Figure 7). Our approach uses real images as training targets and images with removed objects as inputs, along with a complex filtering process to obtain high-quality data. This results in improved consistency where the background remains unchanged.
>
> 2.**Comparisons with Mask-Guided Methods:** Although mask-guided methods like PowerPaint [1] require additional input conditions (e.g., masks), we have demonstrated that Diffree can yield competitive results without such inputs. In Appendix A7, we discuss the challenges of input masks and provide a fair comparison by supplying complete masks for methods that require them. As shown in Figure A10, methods that lack mask information fail to produce satisfactory results.
>
> We provide a comprehensive assessment of Diffree’s performance relative to existing methods through these comparisons.
>
>
>
> [1] Zhuang, Junhao, et al. "A task is worth one word: Learning with task prompts for high-quality versatile image inpainting." ECCV, 2024.
>
> [2] Nichol, Alex, et al. "Glide: Towards photorealistic image generation and editing with text-guided diffusion models." ICML 2022.
>
> [3] Li, Yuheng, et al. "Gligen: Open-set grounded text-to-image generation." CVPR 2023.
>
> [4] Rombach, Robin, et al. "High-resolution image synthesis with latent diffusion models." CVPR 2022.

---

> ### Author Response · Authors · 2024-11-21
> **Response to Reviewer zVNG (Part: 3/3)**
>
> **Q8:** Ambiguity in Object Placement in Complex Scenes: The method struggles with complex images involving multiple entities (e.g., two people). When required to add an object to one specific entity, the model may face ambiguity, as there is limited functionality for users to adjust the mask and specify the target placement.
>
> **A8:** Thank you for your question. As discussed in **A2 (Point 2)**, we retrained Diffree to improve its ability to add new objects through context-related descriptions. For images with multiple entities, Diffree achieves precise object addition by utilizing detailed prompts that specify attributes or positions (e.g., “Add a red knitted wool hat on the head of the doll on the left” in the Figure A14). We believe this effectively addresses your concerns.
>
> **Q9:** How does Diffree handle ambiguities in scenes with multiple entities (e.g., if there are two people in an image, how does it decide which one to add the object to)? Could user input be incorporated to clarify target placement in such cases
>
> **A9:** Please refer to our response in **A8**.
>
> **Q10:** Could more fine-grained prompts be incorporated or generated to enhance the model's ability to handle specific requests for object details and interactions within a scene?
>
> **A10:** Yes. As mentioned in **A2 (Point 2)** and **A8**, Diffree benefits from detailed textual descriptions (e.g., “Add a red knitted wool hat on the head of the doll on the left”) that include specific attributes, locations, and interactions. By incorporating precise language in the prompts, users can guide the model to produce more accurate and contextually appropriate additions.

---

> ### Comment · Reviewer_zVNG · 2024-11-24
>
> Thank you for your response and effort in addressing my concerns. While I appreciate your attempt to clarify the points raised, there are several critical areas where the work still falls short.
>
> ## A2.1
> The attributes presented by the authors fail to convince me of the robustness of the proposed method. The attributes of the objects are overly specific. For example, prompts such as *“glasses with a corgi on the left side and a duck on the right side”* require a level of detail that your method seems ill-equipped to handle. More intricate and realistic prompts of this nature would likely challenge the effectiveness and scalability of your approach.
>
> ## A2.2
> You mentioned re-labeling the dataset using GPT-4. However, the process of re-labeling lacks transparency. How exactly was the re-labeling conducted? How did you ensure objectivity and detail in this process? Additionally, how was this evaluated to confirm its efficacy?
>
> The example provided demonstrates functionality primarily for location attributes, but what happens when users require specific modifications such as:
> - Adjusting the size (*“make it slightly bigger”*): What if the adjustment becomes exaggerated?
> - Changing the shape (*“curve it more elegantly”*): How would your method handle such structural transformations?
> - Altering partial color attributes (e.g., *“change the color of the mirror’s border”*): How is this managed?
>
> The lack of discussion around these nuanced cases is a significant oversight.
>
> ## A5
> I did not specifically mention *"complex interactions"* between objects in my earlier comments. My concern here is that the objects depicted in **Figure 1** appear highly unnatural and unrealistic. They look as though they have been hard-pasted into the scene, lacking any visual cohesion or natural blending. This issue undermines the claims of your method's ability to generate realistic outputs.
>
> ## A6
> The user study appears to be rushed, poorly planned, and subjective. Several key details are missing:
> - **Participant demographics**: Who were the participants? What were their ages, genders, or professions?
> - **Evaluation process**: How was the evaluation conducted? Was the process designed to ensure objectivity?
> - **Setup and methodology**: What specific setup was used for the study? Were participants given clear and standardized tasks?
> - **Evaluation metrics**: What does a "win" or "loss" mean? Is "win" based on realism, retention of background integrity, or some other metric? Without clear definitions, the reported results seem arbitrary and subjective.
>
> Drawing meaningful conclusions from such a study is highly problematic without addressing these issues.
>
> ---
>
> ## Summary
> While I acknowledge the effort made in responding to my concerns during the rebuttal period, the submission still lacks the necessary rigor and detail required to meet the standards of ICLR. The paper shows potential but needs significant refinement. The evaluations, methodologies, and overall presentation require a more meticulous and objective approach.
>
> I am still deliberating on whether to lower the score. My final decision will depend on the perspectives of other reviewers and their comments.

---

> ### Author Response · Authors · 2024-11-24
> **Response to Reviewer zVNG**
>
> We appreciate the reviewers’ thorough feedback on our manuscript. Below are our responses to the specific concerns raised:
>
> 1. **Response to Reviewer‘s A2.1: ill-equipped to handle detail prompts such as “glasses with a corgi on the left side and a duck on the right side”**
>
>    As highlighted in **A2: context-based interactions**, our model is capable of accurately adding objects based on detailed, context-specific descriptions. For instance, **the example “Add a red knitted wool hat on the head of the doll on the left” in Figure A14 demonstrates the model’s ability to handle similar detailed prompts effectively.**
>
> 2. **Response to Reviewer‘s A2.2: users require specific modifications (e.g., "curve it more elegantly")**
>
>    As noted in the opening sentence of our **Abstract**: "this paper addresses an important problem of object addition". **Not for specific modifications.**
>
> 3. **Response to Reviewer‘s A2.2: The process of re-labeling**
>
>    *Regarding re-labeling the dataset‘s details using GPT-4o-mini:**
>
>    We provide the following inputs to GPT-4o-mini for precise re-labeling:
>
>    1. Cropped area image of the object to help GPT-4o-mini understand the object’s attributes.
>    2. Original image containing the object to establish context and correspondence.
>    3. object label description.
>    4. A Prompt of the object to guide the task: *You will be provided with the following: 1.A real image of a scene. 2.A cropped image of a specific object from the scene. 3.The category text of the object (e.g., ‘cup’, ‘chair’). Based on this information, generate a concise description of the object’s appearance and its spatial position in the scene. The description should be no longer than 20 words and focus solely on the object and its immediate spatial relationship. Example: ‘A transparent cup on the table.’ Avoid adding unnecessary context or details beyond the object’s appearance and position.*
>
>    Using these inputs, GPT-4o-mini generates descriptions detailing the specific attributes and relative positional relationships of the objects through structured prompts.
>
> 4. **Response to Reviewer‘s A5:  Figure 1 appear highly unnatural and unrealistic and  not mention "complex interactions" between objects**
>
>    We argue that Figure 1 demonstrates a good level of perspective alignment and consistent reflections between objects and their environment, especially considering the cumulative 16 iterations of text-guided object additions.
>
>    Regarding potential reasons (**"complex interactions" you not mention**) for highly unnatural and unrealistic results you might think of, we analyze that the added interaction areas of multiple objects may lead to a decline in quality, which is an issue faced by **all methods**.
>
> 5. **Response to Reviewer‘s A6:  The user study appears to be rushed, poorly planned, and subjective, which asked for supplemented during the rebuttal period. Lack details of participants's ages, genders, or professions, etc.**
>
>    1. The original paper already includes a user study in the appendix, contrary to your initial comment suggesting its absence.
>
>    2. In response to your reviews of needs for a user study on overall satisfaction, we further conducted additional userstudy as per your feedback. We provide the following details:
>
>       - Participant demographics: Ages: between 20 and 30 years old. Genders:  both men and women. Professions: master’s or doctoral students engaged in Computer Vision related research.
>
>       - Evaluation process: Participants were randomly presented with two images (one from Diffree and one from InstructPix2Pix) in a randomized order. They were asked to evaluate which performed better across various metrics used in our quantitative experiments.
>
>       - Setup and methodology: Evaluations were based on the Unified Metric in our quantitative experiments, which includes criteria such as “Background Consistency,” “Location Reasonableness,” “Object Correlation,” “Object Quality,” and “Diversity.”
>
>       - Evaluation metrics: As indicated in the table caption, we assessed overall satisfaction (i.e., Unified Metric in the experiments section). Only “Win” was marked instead of both “Win” and “Loss”. “Diffree Win” or “InstructPix2Pix Win” means that the user found the results of Diffree or InstructPix2Pix superior. Statistical analysis was performed after participants’ blind evaluations.

---

### Official Review · Reviewer_ethC · 2024-11-02

**Soundness:** 3
**Presentation:** 3
**Contribution:** 3
**Rating:** 6
**Confidence:** 5

**Summary:**

The paper presents a novel model, Diffree, which facilitates the addition of objects to images. The authors have constructed a synthetic dataset named OABench, consisting of 74K real-world tuples, to train Diffree using a Stable Diffusion model augmented with an additional mask prediction module. Diffree predicts the target mask and generates inpainting results simultaneously. Extensive experiments demonstrate Diffree's good performance in adding new objects with high success rates while preserving background consistency, spatial appropriateness, and object relevance and quality.

**Strengths:**

1. Diffree offers a user-friendly approach to inserting objects into images. The mask-free object insertion is particularly useful in practical applications.
2. The creation of OABench, a large-scale synthetic dataset, is a significant contribution, providing a rich resource for training and evaluating object addition models.
3. The OMP module's ability to predict the target mask and generate inpainting results simultaneously is a novel architectural advancement in this field.

**Weaknesses:**

1. [1] proposed a method for mask prediction closely related to Diffree. An in-depth analysis and comparison with this work would be beneficial.
2. All the prompts used in this paper are in the form "add {object}". It is unclear how Diffree generalizes to more precise control, such as "add a dragon in the room".
3. While user-friendly for object insertion, it restricts users from adjusting the mask. In standard image processing, users or designers often need to make adjustments to achieve their desired results.

[1] SmartMask: Context Aware High-Fidelity Mask Generation for Fine-grained Object Insertion and Layout Control

**Questions:**

1. In Fig. 1, the "add dragon" prompt results in "a dragon painting in the frame". Is this behavior as expected?
2. How does Diffree perform with out-of-distribution (OOD) objects not included in OABench?

---

> ### Author Response · Authors · 2024-11-21
> **Response to Reviewer ethC  (Part: 1/2)**
>
> We thank Reviewer ethC for their valuable time and effort in reviewing our paper. We appreciate your thoughtful feedback and have addressed your comments in our revised manuscript. We have summarized the updating of our revision in Summary of Paper Updating and provided an in-depth discussion about our Object Mask Predictor (OMP) module in Genera Response. Below, we respond to each of your concerns in detail.
>
> **Q1:** An in-depth analysis and comparison with SmartMask [1] would be beneficial.
>
> **A1:** Thank you for your suggestion. We agree that a comparison with SmartMask [1] is valuable. SmartMask differs from Diffree in several key aspects, particularly in its reliance on additional models and input requirements during inference:
>
> 1. **Dependency on Additional Models:**
>
>    SmartMask requires two extra models during inference: a panoptic segmentation model [2] and a mask-guided inpainting model [3]. The process involves:
>
>    ​	(1) Converting the input image into a panoptic segmentation map using the segmentation model.
>
>    ​	(2) Using this segmentation map as input to SmartMask to generate the object mask.
>
>    ​	(3) Feeding the mask into the inpainting model to perform object addition.
>
>    Diffree, in contrast, accomplishes object addition using a single model without relying on additional models. This makes Diffree more efficient and less resource-intensive. Moreover, the performance of SmartMask heavily depends on the accuracy of the segmentation and inpainting models, which may introduce limitations due to potential inaccuracies or limited label outputs.
>
> 2. **Input Requirements:**
>
>    SmartMask requires a converted semantic layout map, an object description, and a context description of the final scene to generate the object mask. This reliance on detailed scene context can limit its applicability, especially when such detailed descriptions are unavailable or insufficient.
>
>    Diffree only requires the object description for addition, simplifying the inference process and reducing dependency on additional inputs.
>
> Unfortunately, since SmartMask is not currently open-source, conducting quantitative experimental comparisons is challenging. Nevertheless, we believe Diffree offers significant advantages in terms of efficiency, simplicity, and reduced dependency on external models and detailed context inputs. We have also updated the related work section to include a discussion of SmartMask.
>
>
>
> [1] Singh, Jaskirat, et al. "SmartMask: Context Aware High-Fidelity Mask Generation for Fine-grained Object Insertion and Layout Control." CVPR 2024.
>
> [2] Kirillov, Alexander, et al. "Panoptic segmentation." CVPR 2019.
>
> [3] Zhang, Lvmin, Anyi Rao, and Maneesh Agrawala. "Adding conditional control to text-to-image diffusion models." ICCV 2023.
>
> **Q2:** How Diffree generalizes to more precise control.
>
> **A2:** Thank you for this important question. To enhance precise control in object addition, we extended our model by re-labeling our dataset with accurate location descriptions using GPT-4o-mini and retrained Diffree with these detailed annotations and original annotations.
>
> To address your concerns, we conducted experiments using contextually detailed prompts specifying different locations within the same scene or involving multiple similar objects. As shown in Figure A14, our model can accurately add objects based on context-related descriptions (e.g., “Add a red knitted wool hat on the head of the doll on the left”). This demonstrates that Diffree can be extended to handle more precise control. We plan to further enhance this capability by constructing larger datasets with precise annotations.
>
> **Q3:** While user-friendly for object insertion, it restricts users from adjusting the mask. In standard image processing, users or designers often need to make adjustments to achieve their desired results.
>
> **A3:** We appreciate your concern regarding the ability to adjust masks. Diffree addresses this by generating both the edited image and the object mask through our OMP module. This mask serves as a strong starting point for users and designers who may wish to refine the results. Users can adjust the mask using standard image editing tools or incorporate it into existing mask-guided methods for further refinement.
>
> By providing an initial high-quality mask, Diffree reduces the effort required to obtain an ideal mask, which can be a challenging task when starting from scratch. This feature enhances user control and facilitates a more efficient and flexible editing workflow, aligning with the needs of designers and standard image processing practices.

---

> > ### Comment · Reviewer_ethC · 2024-11-25
> >
> > Thank you for your rebuttal. While most of my concerns have been addressed, there are still a few remaining issues.
> > 1.  GPT-4O Relabeling Details. I would like to know the following details regarding the re-labeling of GPT-4O: such as The prompt used for re-labeling; The time taken for the relabeling process; The cost associated with relabeling the entire dataset; and whether the new model was trained from scratch or from a pre-trained checkpoint.
> > 2. About the "dragon" case. As per my understanding, Stable Diffusion 1.5 has the ability to generate cases like “a dragon flying in a room”. However, I failed to find such a case in the revised version.
> >
> > # Suggestions
> > 1. It is highly recommended to add comparison results based on better pre-trained models such as SDXL. Intuitively, better pre-training generally leads to better results.
> > 2. The precious control ability could potentially be a contribution in the Object Addition area. However, this requires a more thorough evaluation, such as the introduction of new benchmarks and metrics.
> > 3. Regarding the visualization of Figure A4, it is suggested to visualize overlaid images to demonstrate the generated masks. The current form makes it difficult for readers to establish a relationship between the masks and the input image.
> >
> > # Summary
> > I am willing to increase my rating based on the authors' rebuttal. However, it should be noted that there is still room for improvement. In the revised version, I hope the authors can further enhance the evaluation.

---

> > > ### Author Response · Authors · 2024-11-25
> > > **Response to Reviewer ethC**
> > >
> > > Dear Reviewer ethC,
> > >
> > > Thank you for your continued engagement with our work and for sharing your valuable insights and suggestions. We appreciate the opportunity to address your remaining concerns.
> > >
> > > ---
> > >
> > > **Response to Remaining Concerns**
> > >
> > > **Regarding your first concern: GPT-4o-mini Relabeling Details.**
> > >
> > > Thank you for your suggestion. We have updated the paper to include a detailed description of the relabeling process using GPT-4o-mini. We would like to elaborate on the following aspects:
> > >
> > > 1. **relabeling process**
> > >
> > >    We provide the following inputs from our OABench to GPT-4o-mini for precise re-labeling:
> > >
> > >    (1) Cropped area image of the object to help GPT-4o-mini understand the object’s attributes.
> > >
> > >    (2) Original image containing the object to establish context and correspondence.
> > >
> > >    (3) object label description.
> > >
> > >    (4) A Prompt of the object to guide the task: *"You will be provided with the following: 1.A real image of a scene. 2.A cropped image of a specific object from the scene. 3.The category text of the object (e.g., ‘cup’, ‘chair’). Based on this information, generate a concise description of the object’s appearance and its spatial position in the scene. The description should be no longer than 20 words and focus solely on the object and its immediate spatial relationship. Example: ‘A transparent cup on the table.’ Avoid adding unnecessary context or details beyond the object’s appearance and position."*
> > >
> > >    Using these inputs, GPT-4o-mini generates descriptions detailing the specific attributes and relative positional relationships of the objects through structured prompts.
> > >
> > > 2. **The time and the cost for the relabeling process**
> > >
> > >    We used the API provided by OpenAI for relabeling operations, which supports batch processing [1]. This allowed us to complete the relabeling in a short amount of time (less than 3 days). Regarding the cost of using GPT-4o-mini, please refer to OpenAI’s official pricing [2].
> > >
> > > 3. **The new model was trained from a pre-trained SD1.5.**
> > >
> > >    We followed the training process outlined in the paper, with the exception that we used both the original descriptions and the relabeled descriptions as text prompts for training. Therefore, our new model is based on the pre-trained Stable Diffusion 1.5.
> > >
> > > We hope these clarifications are helpful in addressing your concern.
> > >
> > > [1] https://platform.openai.com/docs/guides/batch
> > >
> > > [2] https://openai.com/api/pricing/
> > >
> > > **Regarding your second concern: About the "dragon" case.**
> > >
> > > Thank you for your question. The mention of dragons relates to addressing the concern about out-of-distribution (OOD) objects. Previously, we included an example involving “dragon” in Figure A1, which is the last step of the iterative process of Diffree.
> > >
> > > To better address your concerns, we have updated Figure A14 to include two new examples involving dragons: “Add a black dragon flying in the room” and “Add a brown dragon sitting on the ground” in an anime image. These examples demonstrate our model’s ability to handle OOD objects with more precise control.
> > >
> > > ---
> > >
> > > **Response to Suggestions**
> > >
> > > First, we sincerely thank you for your valuable suggestions, which will help us improve the quality of our paper. We would like to address each of them:
> > >
> > > 1. **Using better pre-trained models such as SDXL**
> > >
> > >    Thank you for your suggestion. Due to resource constraints and to ensure fair comparison with InstructPix2Pix, we used the same pre-trained model (i.e., SD 1.5). We would like to emphasize that in the object addition task, Diffree achieved a 98% success rate, significantly outperforming InstructPix2Pix’s 17%. Our current focus is on improving success rates and consistency in text-guided object addition.
> > >
> > > 2. **Precise Control Ability and the Need for Thorough Evaluation**
> > >
> > >    We appreciate your insight. Our immediate goal is to tackle low success rates and background consistency. We agree that introducing new benchmarks and metrics is necessary for evaluating precise control, which is an interesting future direction.
> > >
> > >    Notably, text-guided methods without explicit context are more challenging than those with context. For example, “Add a cup” vs. “Add a cup on the table”: the former depends entirely on the model’s reasoning for placement, highlighting the increased difficulty.
> > >
> > >    This underscores the challenge of object addition compared to editing/removal, where hints exist in the image. One core metric in our paper is the **reasonableness of the object’s position**
> > >
> > > 3. **Sugestion about the visualization of Figure A4**
> > >
> > >    Thank you for the suggestion. We will update Figure A4 in the final revision to better visualize the generated masks to clearly demonstrate their relationship.
> > >
> > > ---
> > >
> > > Thank you again for your feedback. Your insights help us refine our work and highlight areas for improvement. We are committed to advancing this research and believe addressing these challenges will contribute significantly to the field.
> > >
> > > Best regards,
> > >
> > > Authors of Paper 2788

---

> ### Author Response · Authors · 2024-11-21
> **Response to Reviewer ethC  (Part: 2/2)**
>
> **Q4:** The “add dragon” prompt results in “a dragon painting in the frame.” Is this behavior as expected?
>
> **A4:** We appreciate your observation regarding the “add dragon” prompt resulting in “a dragon painting in the frame”. This behavior is indeed expected and showcases the model’s contextual understanding. Since “dragon” does not exist in our training dataset, Diffree generalizes by integrating the dragon in a contextually appropriate manner. In the setting of a room, adding a dragon as a painting within a frame is a plausible and aesthetically coherent choice. This outcome highlights Diffree’s ability to handle out-of-distribution (OOD) objects by creatively incorporating them into the environment. It demonstrates the model’s generalization capabilities and sensitivity to scene context, ensuring that even unfamiliar objects are added in a way that maintains the coherence and realism of the image.
>
> **Q5:** How does Diffree perform with out-of-distribution (OOD) objects not included in OABench?
>
> **A5:** Diffree demonstrates strong performance when handling out-of-distribution (OOD) objects not included in our OABench dataset. As detailed in the Appendix (Lines 921–922), our model is capable of adding various objects absent from the training data. For example, Figures A1 to A3 illustrate that Diffree can successfully add objects like “dragon”, "necklace" or other OOD items, even in anime-style images, despite being trained on real-world images.
>
> This generalization capability stems from our fine-tuning approach based on the pre-trained SD1.5, which inherently can generate various objects from text descriptions. Therefore, even objects not present in our fine-tuning dataset can be added by Diffree. This reflects Diffree’s robustness and adaptability to different image styles and unseen objects, making it applicable to a wide range of scenarios.

---

> ### Author Response · Authors · 2024-11-24
> **Last three day reminder and looking forward to further discussion**
>
> Dear Reviewer ethC,
>
> Thanks again for your valuable time and insightful comments. We hope that our response can address your concerns. As the deadline for Author/Reviewer discussion period is approaching, we really appreciate if you can let us know whether there still exists any further question about the paper or the response. We are looking forward to further discussion.
>
> Best regards,
>
> Authors of Paper 2788

---

### Official Review · Reviewer_t9nW · 2024-11-03

**Soundness:** 2
**Presentation:** 2
**Contribution:** 2
**Rating:** 5
**Confidence:** 5

**Summary:**

This paper proposes solving the object painting problem with only text as guidance and does not rely on other constraints, e.g., shape mask. The authors first construct a synthetic dataset, called OABench, to facilitate the model’s training with the help of the existing SOTA inpainting model. Then, a diffusion-based model named Diffree is proposed to train on this OABench that can simultaneously output the image with the added object and the object mask. Finally, the authors also conduct comparisons to validate the superiority of their method.

**Strengths:**

- Object inpainting using only text without relying on shape constraints.
- The authors built an OABench to facilitate text-guided object inpainting.
- The results are attractive and the supported applications are interesting.

**Weaknesses:**

- Lack of ablation of the validation model design. For example, what happens to the output if the OMP is removed? BTW, integrating a mask head in the diffusion process is not new, e.g. in [1].
- The comparison is slightly unfair. In Figure-A10, comparing the model you trained on the curated dataset to other methods that were not retrained or fine-tuned is unfair.
- This paper is poorly written, especially the explanation of the charts. For example, in line 50 of the description section, the author provides links to three figures for the reader to view. However, these figures lack sufficient detail to explain what the figure depicts and what the symbols within the figures/captions represent.

[1]. Shadow Generation for Composite Image Using Diffusion Model–CVPR 2024

**Questions:**

- How well does the model generalize? Since the model was trained on a synthetic dataset generated from an inpainting model, it is not surprising that the model may overfit the artifacts introduced by the inpainting model.
- In Line 468, the authors argue that their Local CLIP Score is slightly inferior to the InstructPixel2Pixel due to the latter considers only the successful results are taken into consideration. I wonder what the results will be if you combine both the successful and failure cases.
- Regarding the generalization problem, assuming that a real image contains two dogs, and I first use PowerPaint to delete the two dogs, and then let the proposed Diffree generate one dog, what will the output image show? Will the position of the new dog duplicate the position of the previously removed one?
- Were the compared methods (e.g., InstructPix2Pix) trained on the proposed dataset as well? or just directly copied their pre-trained weights for use?

---

> ### Author Response · Authors · 2024-11-21
> **Response to Reviewer t9nW (Part: 1/2)**
>
> Thank you for your thoughtful review, which has helped us strengthen the manuscript. We have summarized the updating of our revision in Summary of Paper Updating and provided an in-depth discussion about our Object Mask Predictor (OMP) module in Genera Response. Below, we address each of your identified weaknesses:
>
> **Q1:** Lack of ablation of the validation model design, e.g., the output if the OMP is removed.
>
> **A1:** We appreciate your suggestion. In Figure 8, we provided an ablation study on the impact of omitting OMP module in Diffree’s iterative results. Without OMP module, the background deteriorates rapidly after multiple additions, as the model cannot ensure background consistency. This demonstrates the crucial role of the OMP module. Please refer to our General Response about the OMP module for more details.
>
> **Q2:** The comparison in Figure A10 in the Appendix, which does not retrain or fine-tune other methods on our dataset, is unfair.
>
> **A2:** We respectfully disagree for the following reasons:
>
> 1. **Consistency with Prior text-guided methods:** In related text-guided methods [2,3], models fine-tuned with their respective training data are compared without the need for retraining on new datasets.
>
> 2. **Fundamental Differences in Data Requirements:** Mask-guided methods differ from text-guided methods in data requirements. Mask-guided methods can be trained directly on real datasets without synthetic data, as they utilize additional mask conditions.  Specifically, the input image condition can be a real image with the object area removed using the mask, while the same real image (without modifications) serves as the training target. Fine-tuning them on our high-quality, relatively small-scale synthetic dataset, while removing the object area from the inputs, would not improve their performance or alter their input requirements.
>
> 3. **Our Data Generation and usage process are Key Contributions**
>
>    In previous text-guided methods [1,2,3], the input and output both are synthesized images for object addition, which leads to poor consistency and makes it difficult to obtain high-quality data pairs. Creating a high-quality object addition dataset is considered challenging [2].
>
>    Our method uses real images as training targets and images with removed objects as inputs, with a complex filtering process for high-quality data. This approach enhances background consistency in data pairs. Comparing with models trained on their own datasets, like InstructPix2Pix, is fair in this context.
>
>
>
> [1] Brooks, Tim, Aleksander Holynski, and Alexei A. Efros. "Instructpix2pix: Learning to follow image editing instructions." CVPR 2023.
>
> [2] Sheynin, Shelly, et al. "Emu edit: Precise image editing via recognition and generation tasks." CVPR 2024.
>
> [3] Zhang, Kai, et al. "Magicbrush: A manually annotated dataset for instruction-guided image editing." NeurIPS 2024.
>
> **Q3:** The explanation of the charts lack sufficient detail.
>
> **A3:** Thank you for pointing this out. We have updated the paper to enhance clarity, providing more detailed explanations of the charts. Please refer to the updated manuscript for specifics.
>
> **Q4:** Concerns about the generalization performance of the Diffree model due to potential overfitting to artifacts introduced by the inpainting model.
>
> **A4:** We understand your concern but believe Diffree demonstrates strong generalization:
>
> 1. **Output independence from artifacts:** Diffree uses real images as outputs during training, preventing artifacts from affecting the generated results. The added objects naturally integrate with the background, enhancing realism.
>
> 2. **Input robustness:** To address your concerns, we further conduct additional experiments where objects were removed using an inpainting model (e.g., PowerPaint [1]) from an image containing multiple identical objects and then added back using Diffree to see if it would be added onto the same position and size. As shown in Figure A13 of the Appendix, the added objects appeared at different positions and sizes, indicating no overfitting to artifacts and demonstrating generalization.
>
> In addition, our visualizations include real images without artifacts and anime images (Figures A2 and A3 in the Appendix), showcasing Diffree’s adaptability and robustness. Furthermore, we discussed in Lines 921-922 of data collection, and for objects that are not in our dataset, the object addition can also be achieved.
>
>
>
> [1] Zhuang, Junhao, et al. "A task is worth one word: Learning with task prompts for high-quality versatile image inpainting." ECCV 2024.

---

> > ### Comment · Reviewer_t9nW · 2024-11-25
> >
> > Thank you for your rebuttal. After reviewing it, I still have two key questions:
> >
> > 1. It appears that even without the OMP module, the objects (e.g., carpet and birthday cake) can still be generated within reasonable regions. The primary difference seems to be the degradation of texture in the background areas. I believe this could be attributed to noise accumulation during the consecutive generation process. If background replacement is effective after each generation, does this imply that the OMP module is still necessary?
> > 2. Concerning A2, are there any quantitative results available to support this argument?

---

> > > ### Author Response · Authors · 2024-11-26
> > > **Response to Reviewer t9nW (Part 1/2)**
> > >
> > > Dear Reviewer js6X,
> > >
> > > Thank you for your continued engagement with our work and for sharing your valuable insights. We appreciate the opportunity to address your remaining concerns.
> > >
> > >
> > > **Regarding your first question:**
> > >
> > > *"If background replacement is effective after each generation, does this imply that the OMP module is still necessary?"*
> > >
> > > Yes, OMP module remains crucial, particularly for text-guided inpainting methods. While background replacement is indeed effective after each generation, OMP module ensures high-quality and consistent results, as highlighted in the General Response about OMP. To the best of our knowledge, our method is the first to introduce an output mask through the OMP module in text-guided methods. Its integration is a significant contribution of our work, as it enables text-guided object addition without compromising background integrity. Even without the OMP module, our method, trained on our high-quality synthetic dataset, still enables object addition with reasonable positions, sizes, and shapes. However, in addition to noise accumulation, the diffusion model exhibits inconsistencies between the output and input for certain textures and fine details, which affects the output image quality.

---

> > > ### Author Response · Authors · 2024-11-26
> > > **Response to Reviewer t9nW  (Part 2/2)**
> > >
> > > **Regarding your second question:**
> > >
> > > *"Concerning A2, are there any quantitative results available to support this argument of the fairness of Figure A10.?"*
> > >
> > > We appreciate your concern for quantitative evidence. While A2 highlights three points, we are uncertain what specific quantitative experiments could directly address your concerns. We assume you are suggesting fine-tuning other methods using our dataset for comparison. To address this, we outline the following key considerations from a dataset perspective:
> > >
> > > 1. **Training mask-guided methods on our dataset is equivalent to training them on the 74K real images subset of the COCO.** These methods rely on real images as both inputs and targets, not need for synthetic data like our dataset OABench, as shown in the table below. Since data synthesis is not required for mask-guided methods, the dataset they use for training is much larger than that of the text-guided method (e.g., **2.8M of PowerPaint v.s. 74K of Diffree**).
> > >
> > >    - Comparison of Training Data between text-guided methods and mask-guided methods.
> > >
> > >    |                       | Text-guided methods (Diffree, InstructPix2Pix) | Mask-guided methods (PowerPaint, SD Inpaint) |
> > >    | --------------------- | ---------------------------------------------- | -------------------------------------------- |
> > >    | **Input Object Mask** | *[*Not Required*]*                             | Object Mask                                  |
> > >    | **Input Image**       | Synthetic image                                | Real Image x (1 - Object Mask)               |
> > >    | **Output Image**      | Real Image                                     | Real Image                                   |
> > >
> > > 2. **For real images, our dataset (a subset of COCO) does not exhibit higher quality than COCO itself.** The synthesis pipeline of our dataset is optimized for high-quality synthetic images, with no filtering applied to real images.
> > >
> > > 3. **GLIGen [2], evaluated in Figure A10, was trained on COCO, the same dataset as ours.**  Since both our method and GLIGen are trained on COCO, this ensures fair comparison, attributing performance differences to methodological advantages rather than training data discrepancies
> > >
> > > 4. **Fine-tuning for mask-guided methods does not alter their reliance on input masks; therefore, the comparison results remain the same as shown in Figure A10.** Whether or not we use our data for fine-tuning does not affect their requirement for input masks.
> > >
> > > We believe these points support our argument and demonstrate the fairness of our comparative analysis.
> > >
> > >
> > >
> > > **However, if there is still concern with our statement, we are willing to provide quantitative experiments to further address your concerns. We are preparing to fine-tune Mask-guided methods using our dataset to further address your concern.** However, as PowerPaint [1], the primary method we compare against, has not open-sourced its training pipeline, we have decided to fine-tune SD Inpaint [3] instead. This process involves modifying code, implementing fine-tuning, and conducting experimental evaluations, all of which require time. We aim to complete these tasks and provide detailed quantitative results (i.e., main results in Table 1) before the extended discussion period (December 2).
> > >
> > > We would like to emphasize again, as noted in **Point 4**, that the visualization results of the finetuned mask-guided model are expected to remain consistent with those shown in Figure A10. Due to the limited time remaining to modify the PDF (only one day), we regret that we are unable to include these visual results in the revised submission.
> > >
> > > [1] Zhuang, Junhao, et al. "A task is worth one word: Learning with task prompts for high-quality versatile image inpainting." ECCV 2025.
> > >
> > > [2] Li, Yuheng, et al. "Gligen: Open-set grounded text-to-image generation." CVPR 2023.
> > >
> > > [3] Rombach, Robin, et al. "High-resolution image synthesis with latent diffusion models." CVPR 20222.
> > >
> > > ---
> > >
> > > Thank you again for your thoughtful feedback, which continues to refine and strengthen our work. **We look forward to engaging in further discussions to address your concerns comprehensively.**
> > >
> > >
> > >
> > > Best regards,
> > >
> > > Authors of Paper 2788

---

> ### Author Response · Authors · 2024-11-21
> **Response to Reviewer t9nW (Part: 2/2)**
>
> **Q5:** The Local CLIP Score results when combining both successful and failure cases.
>
> **A5:** As detailed in Lines 358–366, the Local CLIP Score requires calculating the CLIP Score for the added object area. For most of InstructPix2Pix’s results, additions fail or alter the entire image, making calculation impossible. Compared to InstructPix2Pix, Diffree has a significantly higher success rate (18% vs. 98%). To ensure fairness, we set the Local CLIP Score of failure cases to zero for both methods. The results highlight Diffree’s advantages:
>
> - The Local CLIP Score results of InstructPix2Pix and Diffree, combining both successful and failure cases. We set the Local CLIP Score of failure cases to zero for both methods.
>
>   | Dataset        | InstructPix2Pix | Diffree |
>   | -------------- | --------------- | ------- |
>   | **COCO**       | 5.10            | 28.53   |
>   | **OpenImages** | 5.52            | 28.23   |
>
>
> In order to better address your concerns, we further conduct a user study on the overall satisfaction with all results between InstructPix2Pix and our Diffree in 1000 cases of COCO and OpenImages. The following comparison demonstrates our significant advantage:
>
> - User study of overall satisfaction in 1000 cases of COCO and OpenImages dataset, with the win rate defined as the ratio of our Diffree wins to the total wins by either Diffree or InstructPix2pix.
>
>   | Dataset        | InstructPix2Pix Win | Tie  | Diffree Win | Win Rate |
>   | -------------- | ------------------- | ---- | ----------- | -------- |
>   | **COCO**       | 13                  | 110  | 877         | 98.5%    |
>   | **OpenImages** | 23                  | 187  | 790         | 97.2%    |
>
>
> These results emphasize Diffree’s significant advantage over InstructPix2Pix.
>
> **Q6:** Generalization problem, assuming that a real image contains two dogs, and I first use PowerPaint to delete the two dogs, and then let the proposed Diffree generate one dog, what will the output image show? Will the position of the new dog duplicate the position of the previously removed one?
>
> **A6:**  Thank you for this insightful question. We further conduct experiments to address this concern (Figure A13 in the Appendix). After removing one object from an image containing multiple identical objects and using Diffree to add it back, we observed that the new object’s position and size generally differ from the removed one. This indicates that Diffree does not duplicate previous positions and demonstrates good generalization ability.
>
> **Q7:** Were the compared methods (e.g., InstructPix2Pix) trained on the proposed dataset as well?
> **A7:** No, InstructPix2Pix and Diffree were trained on their respective datasets based on pre-trained weights of SD 1.5. For fairness concerns, please refer to our detailed discussion in **A2**.

---

> > ### Author Response · Authors · 2024-11-24
> > **Last three day reminder and looking forward to further discussion**
> >
> > Dear Reviewer t9nW,
> >
> > Thanks again for your valuable time and insightful comments. We hope that our response can address your concerns. As the deadline for Author/Reviewer discussion period is approaching, we really appreciate if you can let us know whether there still exists any further question about the paper or the response. We are looking forward to further discussion.
> >
> > Best regards,
> >
> > Authors of Paper 2788

---

> ### Comment · Area_Chair_9okY · 2024-11-25
>
> Dear Reviewer t9nW,
>
> Could you kindly review the rebuttal thoroughly and let us know whether the authors have adequately addressed the issues raised or if you have any further questions.
>
> Best,
>
> AC of Submission2788

---

### Author Response · Authors · 2024-11-21
**Summary of Paper Updating**

We have carefully addressed all reviewers’ comments and concerns. Major modifications are highlighted in blue in the updated manuscript. Below is a summary of the key changes in our updated version:

1. **We added more comparisons and discussion**

- Added in-depth discussion and explanations of the OMP module’s role and training process. (see Section A8 in Appendix).
- Added a user study on the overall satisfaction with all results between InstructPix2Pix and our Diffree. (see Section A9 in Appendix).
- Added discussions and additional experiments to demonstrate generalization and address specific concerns about artifacts. (see Section A10.1 in Appendix).
- Added discussions on the generalization of out-of-distribution (OOD) objects not included in OABench for Diffree. (see Section A10.2 in Appendix).
- Added discussions of the the generation from Diffree regarding the description with specific object attributes and context-based interactions. (see Section A10.3 in Appendix).

2. **We added more experimental results.**

- Explored Diffree's results on the generalization ability for artifacts. (see Figure A13 in Appendix).
- Explored the results regarding users' satisfaction through a user study. (see Figure A12 in Appendix).
- Conducted experiments using contextually detailed prompts specifying different locations within the same scene or involving multiple similar objects. (see Figure A14 in Appendix).
- Explored the results of combining Diffree and shadow generation work. (see Figure A11 in Appendix).

3. **Addressed other phrasing clarifications requested by reviewers.**

---

### Author Response · Authors · 2024-11-21
**General response (About the Object Mask Predictor (OMP) module)**

We sincerely thank all the reviewers for their valuable feedback and suggestions. To address concerns about the Object Mask Predictor (OMP) module in our method, we provide an in-depth discussion summarizing its role:

1. **Ensuring background consistency, which is crucial for iterative additions**

   In mask-guided methods (e.g., PowerPaint), the instance mask is a required input. Typically, a post-processing step involves mix the synthesized object into the input image’s background using the instance mask, ensuring the background remains unchanged.

   In contrast, text-guided methods (e.g., InstructPix2pix and Diffree) do not require an instance mask as input, making this mix operation unavailable. This limitation can lead to quality degradation, especially in iterative additions. To the best of our knowledge, we are the first to introduce an output mask through the OMP module in text-guided inpainting methods. This enables the mix operation and allows for iterative additions. We provided an ablation study (Figure 8 in the paper) to evaluate the impact of omitting the OMP module on Diffree’s iterative results. Without the OMP module, the background deteriorates rapidly after multiple steps, rendering further additions infeasible.

2. **Integration with Existing Works for Exciting Applications**

   As explained in the paper, the instance mask can be used in many applications that require a mask as input. For example, when combined with AnyDoor [1], Diffree can achieve image-prompted object addition. We highlight a few additional points:

   (1) *Combining with shadow generation methods to produce realistic shadows*

   The task, shadow generation, aims to create plausible shadows for a composite foreground, given a composite image without foreground shadows and the foreground object mask [2]. However, existing mask-guided and text-guided inpainting methods pose challenges for shadow generation, especially for objects casting long shadows. These challenges arise for two reasons: first, there is a misalignment between the data masks and the objects and actual shadows (e.g., long shadows); second, for mask-guided method, it is difficult for users to draw the estimated masks of long shadow area with objects , which also is challenging for the model understands the respective parts of shadows and objects in the mask. Therefore, combining inpainting works with shadow generation works can lead to better results.

   In mask-guided methods, the shadow generation method's input can be derived from the input mask and output image. Thanks the OMP module’s output mask, Diffree, as a text-guided method without mask input, effectively integrates with such methods to generate coherent objects and realistic shadows, as shown in Figure A11 of the Appendix.

   (2) *Providing a starting point for user or designer adjustments*

   In standard image processing, users or designers often need to make adjustments to achieve desired results. Diffree’s output mask serves as a good starting point, making it easier for designers to refine the outcome. We emphasize that the OMP’s mask output can be combined with evolving mask-based methods, serving as input for better results or continuous adjustments.

3. **Early Mask Output During Initial Decoding Steps**

   During training, diffusion model takes a concatenation of the input latent $z$ and the noisy output latent $\tilde{z_t}$ to estimate the noise at each timestep. The estimated noise is then used to denoise  $\tilde{z_t}$, producing a noise-free latent $\tilde{o_t}$. The concatenation of $z$ and $\tilde{o_t}$, including input and denoised output information, is passed to OMP to generate the object mask $m$:
   $$
   \tilde{o_t} = \frac{\tilde{z_{t}} - \sqrt{1-\bar{\alpha}t} \epsilon_\theta(\tilde{z{t}}, z, \mathrm{Enc}_{\rm txt}(d), t)}{\sqrt{\bar{\alpha}_t}}
   $$
   This $\tilde{o_t}$ can be derived from the denoising process and is available at each step, enabling mask prediction in the initial steps (see Figure A4 in the Appendix). This allows us to quickly obtain a reasonable mask of the added objects without waiting for complete generation, facilitating integration with various applications.  Training the OMP solely on image pairs and object masks would limit mask generation to after full denoising at inference time.  By aligning the OMP’s training inputs with those of the diffusion model and detaching gradients of $\tilde{o}_t$, we ensure independent optimization without interference, as noted on Line 297. The independence of their loss functions and input consistency make separate training theoretically equivalent to joint training.



[1] Chen, Xi, et al. "Anydoor: Zero-shot object-level image customization." CVPR 2024.

[2] Liu, Qingyang, et al. "Shadow Generation for Composite Image Using Diffusion Model." CVPR 2024.

---

### Meta-Review · Area_Chair_9okY · 2024-12-19

**Metareview:**

(a) This paper introduces Diffree, a diffusion-based model trained on the proposed synthetic OABench dataset for text-guided object painting without additional constraints, demonstrating superior performance through comparisons.

(b) Strengths: The paper presents a novel approach to object inpainting using only text guidance, eliminating the need for shape constraints and manual mask definitions, which significantly enhances usability. The creation of OABench, a large-scale synthetic dataset, is a major contribution, providing a valuable resource for training and evaluating object addition models. Diffree's user-friendly, mask-free object insertion method demonstrates strong practical applicability, supported by its innovative OMP module that predicts masks and generates inpainting results simultaneously. The results are visually appealing, and the well-documented dataset creation process and evaluation metrics ensure clarity and ease of understanding.

(c) Weaknesses: The paper has several weaknesses in the initial version, including a lack of ablation studies for validating model design, and insufficient comparisons with closely related works, particularly methods integrating mask prediction in diffusion models. The evaluation is criticized for unfair comparisons (e.g., using retrained models against non-fine-tuned baselines) and limited diversity in prompts, restricting the model's ability to handle nuanced or context-based interactions. The clarity of methodology descriptions, charts, and figures is lacking, with specific issues in explaining workflows (e.g., Figure 6) and inconsistent visual results (e.g., Figure 1). Additionally, the absence of a comprehensive user study and limited comparative evaluation against more recent inpainting or object addition methods reduce the strength of the paper's claims, while the inability to handle complex scenes or provide mask adjustments limits practical usability.

(d) The most important reasons for reject are: Although this work does show potential, the authors present too many flaws in the paper, which can easily lead to misunderstandings and needs significant refinement. E.g., Figure 1 is intended for stress-test, it is ok but could you please change a case that would not be challenged by the visual consistency issue? How effectively a paper minimizes controversy is also a key measure of its quality. Furthermore, the authors were overly passive during the rebuttal process and should have proactively provided relevant and necessary information in many cases (GPT-4O relabelling, user study, etc.) to convince readers and reviewers. E.g., as for the user study problem, providing the all the details for how to conduct the user study is very important at the first presentation. Although the authors provided additional details in response to the reviewers' further requests, the AC found that it remains unclear how many people participated in the user study and what their relationship is to the 1,000 cases. Were multiple people involved in evaluating a single case to avoid bias?

The AC encourages the authors further minimize controversy and improve the overall quality. However, based on current states and rebuttal/discussion process, the overall evaluation still leans towards rejection. Sorry for this and wish the authors good luck in the next upcoming conference. Your valuable work will eventually be recognized by the community; perhaps it just needs a little more patience.

**Additional Comments On Reviewer Discussion:**

(a) Reviewer t9nW highlights the lack of ablation studies, such as analyzing the impact of removing the OMP module, and notes that integrating a mask head in diffusion models is not novel. They find the comparisons unfair, as the model trained on a curated dataset is compared to methods not retrained or fine-tuned. Additionally, the paper's writing, particularly the explanation of charts and figures, is criticized for being unclear and lacking sufficient detail. The authors have addressed most the concerns, and the reviewer further questions the necessarity of OMP and quantitative support for A2. The first problem seems to be resolved but the second one may needs further discussions.

(b) Reviewer ethC suggests an in-depth analysis and comparison with SmartMask, a method closely related to Diffree for mask prediction. They also raise concerns about the limited prompt structure ("add {object}") and its ability to handle more precise control, like "add a dragon in the room." Additionally, while Diffree is user-friendly for object insertion, it limits users from adjusting the mask, which is often necessary in standard image processing. The authors have addressed most of the concerns at the first time, and further provide more details about GPT-4O relabelling details and the "dragon" case issue.

(c) Reviewer zVNG points out several weaknesses, including the dataset's reliance on generic object labels from COCO, limiting the model's ability to handle more detailed or context-specific prompts. The methodology, particularly the Object Mask Predictor (OMP) module, could benefit from further clarification, and Figure 6 needs a more detailed caption for better understanding. There is visual inconsistency in Figure 1, and the lack of a user study and limited comparative evaluation with other methods reduces the robustness of the findings. The model also struggles with complex scenes involving multiple entities, and the inability to adjust masks for precise object placement is a significant limitation. The authors address parts of the issues, but raise more further concerns during the rebuttal process. The reviewer acknowledges the effort made in responding to the concerns during the rebuttal period. However, he believes the submission still lacks the necessary rigor and detail required to meet the standards of ICLR. The paper shows potential but needs significant refinement. The evaluations, methodologies, and overall presentation require a more meticulous and objective approach.

(d) Reviewer js6X criticizes Diffree for not handling shadows and reflections in object addition and removal, leading to "copy and paste" artifacts, as seen in the "Coconut palm" and "boat" examples. They also point out the lack of discussion and ablation studies regarding the joint training of the OMP and diffusion models, suggesting that OMP could be trained independently. Concerns are raised about Diffree’s generalization, especially in cases like Figure 9(a), where the mask is too small. Additionally, more results are needed to verify if Diffree can handle orientational prompts, such as "on left/right/up/bottom of...". The authors address parts of the concerns.

---

### Decision · Program_Chairs · 2025-01-22

Reject